


# Introduction of the DISAMAR radiative transfer model: Determining Instrument Specifications and Analysing Methods for Atmospheric Retrieval (version 4.1.5)

Johan F. de Haan[1], Ping Wang[1], Maarten Sneep[1], J. Pepijn Veefkind[1], and Piet Stammes[1]

[1]Royal Netherlands Meteorological Institute (KNMI), De Bilt, the Netherlands

**Correspondence:** Ping Wang (ping.wang@knmi.nl)

**Abstract.**

DISAMAR is a computer model developed to simulate retrievals of properties of atmospheric trace gases, aerosols, clouds, and the ground surface from passive remote sensing observations in wavelength range from 270 to 2400 nm. It is being used for the TROPOMI/Sentinel-5P and Sentinel-4/5 missions to derive Level-1b product specifications. DISAMAR uses the doubling-

adding method and the layer-based orders of scattering method for radiative transfer calculations. It can perform retrievals using three different approaches: Optimal Estimation (OE), Differential Optical Absorption Spectroscopy (DOAS), and the combination of DOAS and OE, called DISMAS (DIfferential and SMooth Absorption Separated). The derivatives, which are needed in the OE and DISMAS retrievals, are derived in a semi-analytical way from the adding formulas. DISAMAR uses plane-parallel homogeneous atmospheric layers with a pseudo-spherical correction for large solar zenith angles. DISAMAR has

various novel features and diverse retrieval possibilities such as retrieving aerosol layer height and ozone vertical profile. This paper provides an overview of the DISAMAR model version 4.1.5 without treating all the details. We focus on the principle of the layer-based orders of scattering method, the calculation of the semi-analytical derivatives, and the DISMAS retrieval method, because it is to our knowledge the first time that these methods are described. We demonstrate some applications of DISMAS and the derivatives.

## 1 Introduction

DISAMAR stands for Determining Instrument Specifications and Analysing Methods for Atmospheric Retrieval. It is a computer code written in Fortran 90 to simulate retrievals of atmospheric components like trace gases, aerosols, and clouds, and properties of the ground surface from passive remote sensing observations of the Earth. The wavelength range considered

is from 270 to 2400 nm. The development of DISAMAR started around the year 2000, but made use of the heritage of the doubling-adding codes from the 1980s (De Haan et al., 1987). Initially, DISAMAR has been developed to derive Level-1b radiance requirements given specific Level-2 geophysical product requirements for satellite instruments. In order to derive



Level-1b requirements for a given requirement for a Level-2 product, the model has to simulate instrument features and perform retrievals for the Level-2 product with proper error estimation. From the European satellite missions Sentinel-5p, with TROPOMI on board (Veefkind et al., 2012), Sentinel-4 and Sentinel-5, atmospheric trace gases, cloud, and aerosol products have to be generated from the measured radiance spectra. In principle, it is possible to simulate Level-1b spectra using a radiative transfer (RT) model to modify the simulated spectra with instrument features, like noise, and then run a retrieval algorithm. However, this can introduce inconsistencies between the RT model simulation and the retrieval algorithm, which makes the error estimate of the Level-2 product less accurate. With DISAMAR providing the end-to-end calculation for all the steps from instrument response to retrieved geophysical product, a high degree of internal consistency is achieved. For example, DISAMAR has been used to calculate Level-1b requirements for preparation of the Sentinel-4/5/5p missions and other missions (e.g. De Haan, 2009, 2010, 2012; Sanders et al., 2015; Nanda et al., 2018).

DISAMAR contains a radiative transfer module to simulate the radiance, irradiance, and sun-normalized radiance (or reflectance). The radiative transfer is based on the doubling-adding method and includes a more efficient variant, called Layer-Based Orders of Scattering (LABOS). Polarization (Stokes 4-vectors and $4\times4$ Mueller matrices) and Rotational Raman Scattering (RRS) are included in DISAMAR. The implementation of RRS is based on the publications by Chance and Spurr (1997) and Stam et al. (2002). The derivatives of the radiance or reflectance with respect to the retrieved parameters (also called weighting functions or Jacobians) that are needed for the retrievals are calculated using efficient semi-analytical expressions. The atmosphere is assumed to be plane-parallel with an option for pseudo-spherical correction. Three types of retrievals are available, namely Optimal Estimation (Rodgers, 2000), DOAS (Differential Optical Absorption Spectroscopy) (Platt, 1994), and DISMAS (DIfferential and SMooth Absorption Separated), a version of Optimal Estimation based on a DOAS-like approach. The choice of DOAS and OE retrievals is driven by the Level-2 retrieval algorithms for atmospheric trace gas columns (using DOAS) and ozone profile (using OE).

The atmosphere in DISAMAR is assumed to be in hydrostatic equilibrium, which is used to calculate an altitude grid given a temperature profile and a pressure profile. A cloud or aerosol layer can be modelled by a simple reflecting Lambertian surface, or by a layer of scattering particles with a Henyey-Greenstein phase function or a phase function using expansion coefficients that are read from file (e.g. Mie scattering particles). The surface below the atmosphere is a Lambertian reflector with an albedo that can vary with wavelength. In addition, surface emission can be included to simulate sun-induced fluorescence (Frankenberg et al., 2011).

The applications of DISAMAR are retrievals of the ozone profile from the ultra-violet (UV), the total column of $NO_2$ from the visible (VIS), cloud properties (height, fraction, optical thickness) from the $O_2$ A-band in the near-infrared (NIR), or $CO_2$ from the shortwave-infrared (SWIR). The simulated measurements can be modified by adding noise, offsets to the wavelength scales, and offsets to the radiance and solar irradiance. By evaluating the effects of these modifications on retrieved parameters, requirements on the Level-1b measured spectra can be derived. By changing the *a priori* error of auxiliary data (e.g. surface albedo) the required accuracy of auxiliary data can be determined.

The advantage of DISAMAR is that it includes diverse instrument features, simulations, and retrieval algorithms in one code. For such a complex model, it is not possible to cover all details in one paper. We therefore highlight the novel and unique





features of DISAMAR (version 4.1.5), which are listed in Section 2. Section 3 presents the forward simulations. The retrieval part is described in Section 4. Some applications are shown in Section 5 as examples. Section 6 presents a discussion and
conclusions.

## 2 Novel features in DISAMAR

DISAMAR uses a separate altitude grid for the radiative transfer calculations, which is independent of the grid used for specifying the atmospheric properties. This makes it possible to deal with strong vertical gradients in the radiation field, e.g. near the top of clouds.

The altitude grid, wavelength grid, and polar angles are all defined using Gauss-Legendre division points. This makes the integration more accurate than an equidistant grid with similar number of grid points.

The number of streams used for integration over polar angles when multiple scattering is involved, can be arbitrarily large. Also the number of coefficients used for the expansion of the phase function in Legendre functions can be arbitrarily large. All these features make the DISAMAR calculations more accurate.

DISAMAR provides not just the radiance spectrum of the backscattered sunlight, but also the derivatives with respect to the elements of the state vector. These derivatives are essential in OE and are used to find the solution in an iterative manner. They are also used to determine the error covariance matrix, gain vectors, and the averaging kernel. In DISAMAR all derivatives are calculated in a semi-analytical manner. Generally the calculation of the derivatives takes no more than 10 – 30% of the total calculation time.

Because DISAMAR was originally developed to determine instrument specifications and analyzing methods for atmospheric retrieval, special emphasis is given on the error information of the retrieved products and instrumental features. The main retrieval algorithm is Optimal Estimation (OE) (Rodgers, 2000). Therefore, DISAMAR provides full diagnostic information including error covariance matrices and averaging kernels. It can deal with combined retrievals using information from different wavelength bands. OE is used for ozone profile retrieval by DISAMAR. The retrieved ozone profiles consist of the volume
mixing ratio (VMR) of ozone at each altitude, obtained through spline interpolation or linear interpolation on the logarithm of the VMR given at a user defined altitude grid. This makes conversion from VMR to Dobson Units per layer and from VMR to number densities simple.

By using the newly developed DISMAS approach, which combines the principle of the DOAS retrieval method with Optimal Estimation, the number of wavelengths at which forward calculations have to be performed can be reduced significantly.

## 85 3 DISAMAR forward model

The forward model of DISAMAR provides simulated radiance (or sun-normalized radiance), irradiance, and the derivatives of the radiance with respect to the state vector elements. In this section attention is given to the calculation of these derivatives using a semi-analytical approach, because this important aspect for retrievals has not been published before.





### 3.1 Instrument model

The simulated radiance and irradiance can be convoluted separately with the instrument spectral response function (ISRF). The ISRF is defined in terms of a convolution operator. A Gaussian ISRF and a flat-topped ISRF are included in DISAMAR. A tabulated ISRF, e.g. measured for a specific instrument, can also be used as an external file. The instrument features such as Signal to Noise Ratio (SNR), stray light, and wavelength calibration errors can be simulated using DISAMAR by specifying corresponding entries in a configuration file.

### 95 3.2 Atmospheric and surface properties

#### 3.2.1 Gas absorptions

Gas absorptions include those from line absorbers and non-line absorbers. In the wavelength region considered here (270 nm to 2400 nm) line absorbers are $H_2O$, $CO_2$, $CO$, $CH_4$, and $O_2$. Non-line absorbers are $O_3$, $SO_2$, $O_2$-$O_2$, $BrO$, $HCHO$, $NO_2$, and $CHOCHO$. The collision complexes, such as $O_2$-$O_2$ and $N_2$-$O_2$, absorb light at UV, VIS, NIR and SWIR (shortwave

infrared) wavelengths. The amount of absorption by the collision complexes generally scales with the square of the pressure. In DISAMAR absorption by $N_2$-$O_2$ (in the spectral region of the $O_2$ A-band) and the absorption by $O_2$-$O_2$ are simulated.

The standard database for line absorbing molecules is the HITRAN 2008 database (Rothman et al., 2009). Line parameters are read from the HITRAN database for a particular gas and a Voigt profile is used to calculate the absorption cross-section. For $H_2O$, $CO_2$, $CO$, and $CH_4$ line mixing is ignored. For $O_2$ line mixing is taken into account using the model described in

Tran and Hartmann (2008) for the $O_2$ A-band (De Haan, 2012).

The gas absorptions from line absorbers have to be calculated line-by-line at a high-resolution wavelength grid, and then convolved with the ISRF. Gaussian division points are used to define the wavelength grid, which improves the accuracy of the integration during the convolution of the ISRF. For constructing the wavelength grid, we start from the shortest wavelength with the interval of one full-width-half-maximum (FWHM) of the ISRF. If there is a strong absorption line in the interval, the

interval is reduced until the position of the strong line is reached, so this interval is smaller than the FWHM. The next interval starts from the position of the strong line. This procedure is repeated until the end of the wavelength range. The number of Gaussian points is scaled with the size of the interval, which means that a smaller interval is having less Gaussian points than the number of Gaussian points specified for the FWHM. Therefore, the wavelength grid is not equidistant: a finer wavelength grid is used for denser absorption lines, and a coarser grid is created if there are no absorption lines. Typically the number

of Gaussian division points is between 3 and 30 per wavelength interval in the $O_2$ A-band for a FWHM of 0.5 nm. Figure 2 shows two 1-nm wide parts of the $O_2$ A-band spectrum, from 757.8 to 758.8 nm and from 765.2 to 766.2 nm, simulated using DISAMAR. The used spectral resolution (wavelength difference between two grid points) is also shown in Fig. 2. In the spectral part from 757.8 to 758.8 nm, which is devoid of $O_2$ absorption lines, the wavelength intervals are 0.4 nm, and every interval is divided into 40 Gaussian points (see Fig. 2 (a, b)). In the spectral part from 765.2 to 766.2 nm, which has many $O_2$

absorption lines, the intervals are smaller due to the absorption lines (see Fig. 2 (c, d)).





Vertical profiles of absorbing gases are specified using the volume mixing ratio (VMR, in parts per million by volume, ppmv) at pressure grid levels. The volume mixing ratio has the advantage that it remains constant if the temperature changes. The number density of gas molecules is calculated internally using hydrostatic equilibrium. The pressure profile and temperature profile grid can be different for simulation and retrieval: for example, trace gas profile retrievals may be using 50 grid points
for the simulation to have high forward model accuracy, but only 12 or 18 grid points for the retrieval, to represent the limited information content of the measurements.

### 3.2.2 Cloud and aerosol layers

Two different types of cloud and aerosol layers are distinguished in DISAMAR: an opaque Lambertian reflector and a layer of scattering particles. For the Lambertian type, the cloud/aerosol reflector is located at the top of a pressure interval. For the
scattering particles layer, a Henyey-Greenstein phase function or a Mie phase function (or phase matrix in case of polarization) can be used for aerosols and clouds in DISAMAR. If the Henyey-Greenstein phase function is used, the wavelength dependence of the optical thickness is modeled using the Ångstrøm exponent and the absorption is controlled by the single scattering albedo of the aerosol and cloud particles. A Mie phase function is provided in the form of expansion coefficients (De Rooij and Van der Stap, 1984) in an external file. It is assumed that in general a part of the pixel is cloud-free and a part is covered by cloud.
This part is controlled by the cloud fraction, which can have values in the interval [0,1].

In DISAMAR the atmosphere is vertically divided into pressure intervals. The surface pressure (say 1000 hPa) defines the lower boundary of the first interval. The intervals are further specified by the pressures of the top of the intervals. For example, if the pressure levels are specified at 700, 600, and 0.1 hPa, the atmosphere has 3 intervals: from the surface to 700 hPa, from 700 to 600 hPa, and from 600 to 0.1 hPa. The top of the atmosphere (TOA) is at 0.1 hPa. This division is suited to model a cloud
layer with boundaries at 700 and 600 hPa. Internally the pressure levels are translated into altitude levels assuming hydrostatic equilibrium. For this calculation, the temperatures at the pressure levels should be provided in the configuration file.

In order to perform radiative transfer calculations, each interval is divided into a number of homogeneous layers using Gaussian division points for the altitude. For example, the three intervals from the surface to TOA may be divided into layers using 12, 10, 28 Gaussian division points, respectively. The standard atmospheres can be used as input for the pressure,
temperature, and gas mixing ratio profiles.

Figure 3 illustrates the pressure intervals and the layers in the intervals used in DISAMAR. It is important to use a sufficient number of Gaussian division points for each pressure interval. Actually each layer is further divided into sub-layers using Gaussian division points, with typically 4 sub-layers per layer. In this way an accurate integration over altitude can be performed in order to get the layer quantities. Averaging of the optical properties and radiation quantities at the top and bottom of a layer
is not sufficiently accurate if the optical properties differ strongly between top and bottom of the layer. The VMR profiles of gases are interpolated at the radiative transfer model grid. For optically thick clouds we suggest to use 15 – 50 Gaussian division points for the cloud layer, in particular if the cloud parameters for that layer are to be fitted. For optically thin clouds, 10 Gaussian division points should be sufficient for the cloud layer. We use Gaussian division points, because it improves the accuracy in the integration over the altitude.



### 3.2.3 Surface albedo

The surface below the atmosphere is a Lambertian, i.e. isotropically reflecting, surface. In DISAMAR one can choose between a wavelength-independent surface albedo or a surface albedo that has a polynomial wavelength dependence for each spectral band.

Sun-induced fluorescence from vegetation due to photosynthesis can be observed in the near-infrared (Joiner et al., 2011; Frankenberg et al., 2011). In order to be able to fit fluorescence in DISAMAR, a polynomial describing surface emission can be used in combination with a wavelength-dependent surface albedo.

### 3.3 Radiative transfer

### 3.3.1 Doubling-adding method

The doubling-and-adding method (also called adding method) is described in many books and papers on radiative transfer in planetary atmospheres (e.g. Van de Hulst, 1981; Hansen and Travis, 1974; Hovenier et al., 2004). Here we use the notation that follows De Haan et al. (1987) and Hovenier et al. (2004). We present the new formulas needed to calculate the derivatives; for the detailed formulas used in the doubling-adding method we refer to De Haan et al. (1987) and Hovenier et al. (2004).

DISAMAR uses the doubling method for the calculation of the scattering properties of the individual atmospheric layers. The adding method is used to add different atmospheric layers, and thus construct the entire atmosphere. As discussed by De Haan et al. (1987) and Stammes et al. (1989) the doubling-adding method can be used to calculate the internal radiation field at the interface between layers in an efficient manner using principles of invariance. This approach can be extended to obtain the derivatives of the internal radiation field, in a linearization step, with respect to a parameter of the atmosphere-surface system. Because the adding formulas are needed in the derivation of the linearization, we provide them here as well.

Radiative transfer matrices are denoted in bold font, and they operate on incident radiation from right to left. A beam of polarized radiation is represented by the Stokes 4-vector $\mathbf{I} = [I, Q, U, V]^T$. Incident sunlight is the Stokes 4-vector $\mathbf{E_0} = [E_0, 0, 0, 0]^T$, where $E_0$ is the extraterrestrial solar irradiance. The elements of the 4×4 scattering matrix $\mathbf{F}$ are expanded into generalized spherical functions, where the expansion coefficients are calculated using the Mie computer program (De Rooij and Van der Stap, 1984). A generalized form of the addition theorem is used to calculate the Fourier coefficients of the 4×4 phase matrix, denoted by $\mathbf{Z}$. $\mathbf{Z}$ is found by rotating $\mathbf{F}$ to the local meridian plane of incident and scattered light, respectively. Figure 1 illustrates the definition of the directions used in DISAMAR.

### 3.3.2 Radiation quantities for an optically thin layer

The doubling method is started with an optically thin layer, with an optical thickness of e.g. $1 \times 10^{-5}$. For a thin layer the Fourier coefficients (indicated by $m$) of the reflection matrix $\mathbf{R}^m$ and the transmission matrix $\mathbf{T}^m$ of the layer illuminated at





the top, are given by:

$$\mathbf{R}_{\text{thin}}^m(\mu,\mu') = dz \frac{k_{\text{sca}}(z)}{4\mu\mu'} \mathbf{Z}^m(-\mu,\mu') \tag{1}$$

$$\mathbf{T}_{\text{thin}}^m(\mu,\mu') = dz \frac{k_{\text{sca}}(z)}{4\mu\mu'} \mathbf{Z}^m(\mu,\mu'). \tag{2}$$

If the thin layer is illuminated at the bottom, the reflection and transmission matrix Fourier coefficients are given by:

$$\mathbf{R}_{\text{thin}}^{*m}(\mu,\mu') = dz \frac{k_{\text{sca}}(z)}{4\mu\mu'} \mathbf{Z}^m(\mu,-\mu') \tag{3}$$

$$\mathbf{T}_{\text{thin}}^{*m}(\mu,\mu') = dz \frac{k_{\text{sca}}(z)}{4\mu\mu'} \mathbf{Z}^m(-\mu,-\mu'). \tag{4}$$

Here $k_{\text{sca}}$ is the volume scattering coefficient of the layer, $z$ is the altitude, and $\mu$, $\mu'$ are the cosines of the polar angles for scattered and incident light, respectively. The '*' indicates quantities with illumination at the bottom. $\mathbf{Z}^m(\pm\mu,\pm\mu')$ are the Fourier coefficients of the phase matrix of the layer. We assume that the scattering matrix does not depend on altitude within a homogeneous layer. The unit of $k_{\text{sca}}$ is $\text{km}^{-1}$ if $z$ is in km.

Attenuation of an incident beam of light due to extinction (sum of scattering and absorption) by the thin layer is given by the direct transmission matrix $\mathbf{E}_{\text{thin}}$:

$$\mathbf{E}_{\text{thin}}(\mu,\mu') = \left(1 - \frac{k_{\text{ext}}(z)}{\mu} dz\right) \delta(\mu - \mu') \, \mathbf{1} \tag{5}$$

where the volume extinction coefficient $k_{\text{ext}}(z) = k_{\text{sca}}(z) + k_{\text{abs}}(z)$, with $k_{\text{abs}}(z)$ the volume absorption coefficient. $\delta(\mu - \mu')$ is the Dirac delta function and $\mathbf{1}$ is the unity matrix. Equation 5 follows from the Lambert-Beer attenuation law by keeping only the linear term of a Taylor series expansion.

### 3.3.3 Adding method formulae

We now consider adding two layers on top of each other. Let $\mathbf{E}_k$, $\mathbf{T}_k$, and $\mathbf{R}_k$ be the matrices representing the Fourier coefficients of the direct transmission, diffuse transmission, and reflection for illumination at the top of layer $k$, respectively. Here we omitted the Fourier index $m$ for brevity. Similarly, let $\mathbf{E}_k^*$, $\mathbf{T}_k^*$, and $\mathbf{R}_k^*$ be these matrices for illumination at the bottom of layer $k$. The matrices for layer $k+1$ are denoted similarly. $\mathbf{E}_k$ is a diagonal matrix. For a plane-parallel atmosphere its elements are given by

$$E_{4(i-1)+j,4(i-1)+j} = \exp(-\tau/\mu_i) \tag{6}$$

where $i$ is the index of the Gaussian $\mu$-point, $j = 1,2,3,4$ is the Stokes parameter index, and $\tau$ is the layer optical thickness.

The adding scheme for illumination at the top of the two layers is presented in Eqs. 7–12. The adding scheme for illumination at the bottom of the two layers is presented in Appendix B. The adding scheme is used for every Fourier coefficient; at the end the reflection and transmission matrices are obtained by summing up the Fourier coefficients.





The scheme of adding layer $k+1$ on top of layer $k$ to create a combined layer is illustrated in Figure 4. The adding method formulae are:

$$\mathbf{E}_{k,k+1} = \mathbf{E}_k \mathbf{E}_{k+1} \tag{7}$$

$$\mathbf{Q}_{k,k+1} = \mathbf{R}_{k+1}^* \mathbf{R}_k + \mathbf{R}_{k+1}^* \mathbf{R}_k \mathbf{R}_{k+1}^* \mathbf{R}_k + ... \tag{8}$$

$$\mathbf{D}_{k,k+1} = \mathbf{T}_{k+1} + \mathbf{Q}_{k,k+1} \mathbf{E}_{k+1} + \mathbf{Q}_{k,k+1} \mathbf{T}_{k+1} \tag{9}$$

$$\mathbf{U}_{k,k+1} = \mathbf{R}_k \mathbf{E}_{k+1} + \mathbf{R}_k \mathbf{D}_{k,k+1} \tag{10}$$

$$\mathbf{R}_{k,k+1} = \mathbf{R}_{k+1} + \mathbf{E}_{k+1} \mathbf{U}_{k,k+1} + \mathbf{T}_{k+1}^* \mathbf{U}_{k,k+1} \tag{11}$$

$$\mathbf{T}_{k,k+1} = \mathbf{T}_k \mathbf{E}_{k+1} + \mathbf{T}_k \mathbf{D}_{k,k+1} + \mathbf{E}_k \mathbf{D}_{k,k+1}. \tag{12}$$

Here $\mathbf{E}_{k,k+1}$ is the attenuation of incident light by the combined layer. $\mathbf{Q}_{k,k+1}$ is the sum of the repeated reflections between
the layers for a unit amount of light incident on the interface between the layers. $\mathbf{D}_{k,k+1}$ is the diffuse downward light at the interface between the layers, $\mathbf{U}_{k,k+1}$ is the upward light at the interface between the layers, $\mathbf{R}_{k,k+1}$ is the reflectance of the combined layer, and $\mathbf{T}_{k,k+1}$ is the transmission of the combined layer.

The matrices $\mathbf{R}$, $\mathbf{T}$, $\mathbf{E}$, $\mathbf{Q}$, $\mathbf{D}$, and $\mathbf{U}$ are so-called supermatrices. The elements of supermatrices contain Gaussian points for the polar angle with weights that are suited for numerical integration by multiplication of two supermatrices. For the
definition of supermatrices, we refer to De Haan et al. (1987). The total number of polar angles, $N_\mu$, is the number of Gaussian division points, $N_g$, plus one solar zenith angle and one viewing zenith angle, pertaining to the selected sun-view geometry. The dimension of the supermatrix is $(N_\mu, N_\mu)$ if polarization is ignored and $(4N_\mu, 4N_\mu)$ if polarization is included using all 4 Stokes parameters. In the following a supermatrix is meant when we speak of a matrix.

The formula for repeated reflections at the interface, Eq. 8, can also be written as:

$$\mathbf{1} + \mathbf{Q}_{k,k+1} = (\mathbf{1} - \mathbf{R}_{k+1}^* \mathbf{R}_k)^{-1} \tag{13}$$

where $\mathbf{1}$ is the unity matrix. Inserting Eq. 13 into Eq. 9 yields:

$$(\mathbf{1} - \mathbf{R}_{k+1}^* \mathbf{R}_k)(\mathbf{E}_{k+1} + \mathbf{D}_{k,k+1}) = \mathbf{E}_{k+1} + \mathbf{T}_{k,k+1}. \tag{14}$$

Equations 13 and 14 are needed in the derivation of the derivatives, discussed next.

### 3.3.4 Derivatives of the reflectance using the adding method

Retrieval algorithms that are designed to derive quantitative information about properties of the atmosphere from measured spectra do not only require an atmospheric model to simulate reflectance spectra, but also derivatives of the reflectance with respect to the optical properties of the atmosphere – sometimes called Jacobians or weighting functions (see e.g. Spurr (2006); Hasekamp and Landgraf (2005); Bai et al. (2020); Rodgers (2000)). Obviously, these derivatives can be calculated using numerical differentiation based on perturbation. However, numerical differentiation is a slow process, in particular when altitude
profiles have to be determined and the atmosphere consists of 20 – 100 layers. Here we derive semi-analytical expressions for





the derivatives of the reflectance with respect to the optical properties of the atmospheric layers. The semi-analytical derivatives are calculated at the interfaces of the layers, i.e at levels. The derivatives for the layers themselves can then be calculated from the derivatives at the interfaces using interpolation and integration over the layers. Note that often the derivatives are calculated pertaining to optical properties of atmospheric layers, which differs from our approach.

The derivatives can be found by calculating the change of reflectance at top-of-atmosphere by a change of atmospheric properties at each height, $d\mathbf{R}/dz$. In the adding scheme, such a change can be invoked by adding a thin layer at altitude $z$. As shown by Eqs. 1 - 5 the reflection and transmission of a thin layer can be written analytically as a function of the optical properties of the layer: $k_{\mathrm{sca}}$, $k_{\mathrm{abs}}$, and $\mathbf{Z}$. Therefore, the computation of semi-analytical derivatives should involve a thin layer. If one first calculates the reflection of two layers on top of each other, $\mathbf{R}_{\mathrm{top+bot}}$, and next calculates the reflection of the two layers

with a thin layer in between, $\mathbf{R}_{\mathrm{top+thin+bot}}$, then $d\mathbf{R}/dz$ can be calculated from the differential $(\mathbf{R}_{\mathrm{top+thin+bot}} - \mathbf{R}_{\mathrm{top+bot}})/dz$.

We divide the atmosphere into two parts, a top part and a bottom part. Let $\mathbf{R}_{\mathrm{top}}$ , $\mathbf{R}_{\mathrm{top}}^{*}$ , $\mathbf{T}_{\mathrm{top}}$, $\mathbf{T}_{\mathrm{top}}^{*}$ be the Fourier coefficients of the reflection and transmission matrices of the top part; the quantities without an asterisk denote the properties for illumination at the top, the quantities with an asterisk denotes illumination at the bottom. Similarly, matrices with the subscript 'bot' denote the properties of the lower partial atmosphere.

Using the adding formulae, Eqs. 7–12, and keeping only constant terms and terms linear in $dz$, one can derive the derivative $d\mathbf{R}/dz$ in a concise form as:

$$\frac{d\mathbf{R}}{dz} = [\mathbf{E}_{\mathrm{top}} + \mathbf{T}_{\mathrm{top}}^{*}](\mathbf{1} + \mathbf{Q}^{*})\mathbf{W}(z)(\mathbf{1} + \mathbf{Q})[\mathbf{E}_{\mathrm{top}} + \mathbf{T}_{\mathrm{top}}] \tag{15}$$

where $\mathbf{Q}$ and $\mathbf{Q}^{*}$ are given by:

$$\mathbf{Q} = \mathbf{R}_{\mathrm{top}}^{*}\mathbf{R}_{\mathrm{bot}} + \mathbf{R}_{\mathrm{top}}^{*}\mathbf{R}_{\mathrm{bot}}\mathbf{R}_{\mathrm{top}}^{*}\mathbf{R}_{\mathrm{bot}} + ... \tag{16}$$

$$\mathbf{Q}^{*} = \mathbf{R}_{\mathrm{bot}}\mathbf{R}_{\mathrm{top}}^{*} + \mathbf{R}_{\mathrm{bot}}\mathbf{R}_{\mathrm{top}}^{*}\mathbf{R}_{\mathrm{bot}}\mathbf{R}_{\mathrm{top}}^{*} + ... \tag{17}$$

and $\mathbf{W}(z)$ is given by:

$$\mathbf{W}(z) = k_{\mathrm{sca}}(z)\mathbf{Z}_{-+}'(z) + \left[k_{\mathrm{sca}}(z)\mathbf{Z}_{--}'(z) - k_{\mathrm{ext}}(z)[\mathbf{1}/\boldsymbol{\mu}]\right]\mathbf{R}_{\mathrm{bot}}+$$
$$\mathbf{R}_{\mathrm{bot}}\left[k_{\mathrm{sca}}(z)\mathbf{Z}_{++}'(z) - k_{\mathrm{ext}}(z)[\mathbf{1}/\boldsymbol{\mu}]\right] + \mathbf{R}_{\mathrm{bot}}\left[k_{\mathrm{sca}}(z)\mathbf{Z}_{+-}'(z)\right]\mathbf{R}_{\mathrm{bot}}. \tag{18}$$

Here $\mathbf{1}/\boldsymbol{\mu}$ is the supermatrix with the elements $1/\mu$ on the diagonal, representing the slant path for extinction. $\mathbf{Z}_{\pm\pm}'$ are

abbreviated notations of $\mathbf{Z}'(\pm\mu, \pm\mu')$, the supermatrices of the Fourier coefficients of the phase matrix. The full derivation of Eq. 15 is given in Appendix A.





Inserting Eq. 18 into Eq. 15 yields the separate terms of the derivative:

$$\frac{d\mathbf{R}}{dz} = k_{\text{sca}}(z)[\mathbf{E}_{\text{top}} + \mathbf{T}^*_{\text{top}}](\mathbf{1} + \mathbf{Q}^*)\mathbf{Z}'_{-+}(z)(\mathbf{1} + \mathbf{Q})[\mathbf{E}_{\text{top}} + \mathbf{T}_{\text{top}}]$$

$$+ k_{\text{sca}}(z)[\mathbf{E}_{\text{top}} + \mathbf{T}^*_{\text{top}}](\mathbf{1} + \mathbf{Q}^*)\mathbf{Z}'_{--}(z)\mathbf{R}_{\text{bot}}(\mathbf{1} + \mathbf{Q})[\mathbf{E}_{\text{top}} + \mathbf{T}_{\text{top}}]$$

$$+ k_{\text{sca}}(z)[\mathbf{E}_{\text{top}} + \mathbf{T}^*_{\text{top}}](\mathbf{1} + \mathbf{Q}^*)\mathbf{R}_{\text{bot}}\mathbf{Z}'_{++}(z)(\mathbf{1} + \mathbf{Q})[\mathbf{E}_{\text{top}} + \mathbf{T}_{\text{top}}]$$

$$+ k_{\text{sca}}(z)[\mathbf{E}_{\text{top}} + \mathbf{T}^*_{\text{top}}](\mathbf{1} + \mathbf{Q}^*)\mathbf{R}_{\text{bot}}\mathbf{Z}'_{+-}(z)(\mathbf{1} + \mathbf{Q})[\mathbf{E}_{\text{top}} + \mathbf{T}_{\text{top}}]$$

$$- k_{\text{ext}}(z)[\mathbf{E}_{\text{top}} + \mathbf{T}^*_{\text{top}}](\mathbf{1} + \mathbf{Q}^*)[\mathbf{1}/\boldsymbol{\mu}]\mathbf{R}_{\text{bot}}(\mathbf{1} + \mathbf{Q})[\mathbf{E}_{\text{top}} + \mathbf{T}_{\text{top}}]$$

$$- k_{\text{ext}}(z)[\mathbf{E}_{\text{top}} + \mathbf{T}^*_{\text{top}}](\mathbf{1} + \mathbf{Q}^*)\mathbf{R}_{\text{bot}}[\mathbf{1}/\boldsymbol{\mu}](\mathbf{1} + \mathbf{Q})[\mathbf{E}_{\text{top}} + \mathbf{T}_{\text{top}}]. \tag{19}$$

Since we want to calculate the derivative $d\mathbf{R}/dz$ with respect to the scattering and absorption properties of the atmosphere at altitude $z$, Eq. 19 has to be expressed in the internal fields $\mathbf{D}$ and $\mathbf{U}$ at altitude $z$. Therefore, in Eqs. 20 – 23 the internal fields are expressed using $(\mathbf{1} + \mathbf{Q})$ or $(\mathbf{1} + \mathbf{Q}^*)$ and $\mathbf{R}$, $\mathbf{T}$, and $\mathbf{E}$. The relations are:

$$\mathbf{E}_{\text{top}} + \mathbf{D} = (\mathbf{1} + \mathbf{Q})(\mathbf{E}_{\text{top}} + \mathbf{T}_{\text{top}}) \tag{20}$$

$$\mathbf{U} = \mathbf{R}_{\text{bot}}(\mathbf{1} + \mathbf{Q})(\mathbf{E}_{\text{top}} + \mathbf{T}_{\text{top}}) \tag{21}$$

$$\mathbf{E}_{\text{top}} + \Delta_3\tilde{\mathbf{D}} = (\mathbf{E}_{\text{top}} + \tilde{\mathbf{T}}_{\text{top}})(\mathbf{1} + \tilde{\mathbf{Q}}) \tag{22}$$

$$\Delta_3\tilde{\mathbf{U}}\Delta_3 = (\mathbf{E}_{\text{top}} + \mathbf{T}^*_{\text{top}})(\mathbf{1} + \mathbf{Q}^*)\mathbf{R}_{\text{bot}}. \tag{23}$$

Here ˜ indicates the transpose of a supermatrix and $\Delta_3 = \text{diag}(1, 1, -1, 1)$ is a diagonal matrix which occurs due to polarization. By replacing the $(\mathbf{1} + \mathbf{Q})$ and $(\mathbf{1} + \mathbf{Q}^*)$ related terms in Eq. 19 by the relations in Eqs. 20 - 23, we obtain the derivative in terms of the internal radiation fields:

$$\frac{d\mathbf{R}}{dz} = k_{\text{sca}}(z)[\mathbf{E}_{\text{top}} + \Delta_3\tilde{\mathbf{D}}\Delta_3]\mathbf{Z}'_{-+}(z)[\mathbf{E}_{\text{top}} + \mathbf{D}]$$

$$+ k_{\text{sca}}(z)[\mathbf{E}_{\text{top}} + \Delta_3\tilde{\mathbf{D}}\Delta_3]\mathbf{Z}'_{--}(z)\mathbf{U}$$

$$+ k_{\text{sca}}(z)\Delta_3\tilde{\mathbf{U}}\Delta_3\mathbf{Z}'_{++}(z)[\mathbf{E}_{\text{top}} + \mathbf{D}]$$

$$+ k_{\text{sca}}(z)\Delta_3\tilde{\mathbf{U}}\Delta_3\mathbf{Z}'_{+-}(z)\mathbf{U}$$

$$- k_{\text{ext}}(z)[\mathbf{E}_{\text{top}} + \Delta_3\tilde{\mathbf{D}}\Delta_3][\mathbf{1}/\boldsymbol{\mu}]\mathbf{U}$$

$$- k_{\text{ext}}(z)\Delta_3\tilde{\mathbf{U}}\Delta_3[\mathbf{1}/\boldsymbol{\mu}][\mathbf{E}_{\text{top}} + \mathbf{D}]. \tag{24}$$

Using Eq. 24, we find the partial derivative of the reflectance with respect to the volume absorption coefficient (Eq. 25) and the partial derivative of the reflectance with respect to the volume scattering coefficient (Eq. 26):

$$\frac{\partial^2\mathbf{R}}{\partial k_{\text{abs}}\partial z} = -[\mathbf{E}_{\text{top}} + \Delta_3\tilde{\mathbf{D}}\Delta_3][\mathbf{1}/\boldsymbol{\mu}]\mathbf{U} - \Delta_3\tilde{\mathbf{U}}\Delta_3[\mathbf{1}/\boldsymbol{\mu}][\mathbf{E}_{\text{top}} + \mathbf{D}] \tag{25}$$






$$
\begin{aligned}
\frac{\partial^2 \mathbf{R}}{\partial k_{\text{sca}} \partial z} = {} & [\mathbf{E}_{\text{top}} + \Delta_3 \tilde{\mathbf{D}} \Delta_3] \mathbf{Z}'_{-+}(z)[\mathbf{E}_{\text{top}} + \mathbf{D}] \\
& + [\mathbf{E}_{\text{top}} + \Delta_3 \tilde{\mathbf{D}} \Delta_3] \mathbf{Z}'_{--}(z)\mathbf{U} \\
& + \Delta_3 \tilde{\mathbf{U}} \Delta_3 \mathbf{Z}'_{++}(z)[\mathbf{E}_{\text{top}} + \mathbf{D}] \\
& + \Delta_3 \tilde{\mathbf{U}} \Delta_3 \mathbf{Z}'_{+-}(z)\mathbf{U} \\
& - [\mathbf{E}_{\text{top}} + \Delta_3 \tilde{\mathbf{D}} \Delta_3][\mathbf{1}/\boldsymbol{\mu}]\mathbf{U} \\
& - \Delta_3 \tilde{\mathbf{U}} \Delta_3 [\mathbf{1}/\boldsymbol{\mu}][\mathbf{E}_{\text{top}} + \mathbf{D}].
\end{aligned}
\tag{26}
$$

The above relation, Eq. 26, is a generic expression for the derivative with respect to the volume scattering coefficient. However, since it contains the phase matrix it makes a difference whether the change in the volume scattering coefficient is due to a
change in the number of scattering molecules in the thin layer, or due to a change in the number of aerosol particles. If the number of scattering molecules is changing, the phase matrix of the molecules has to be used in Eq. 26, whereas the phase matrix of the aerosol particles has to be used if the number of aerosol particles is changing. In general one has to distinguish between molecules, aerosols, and cloud particles.

### 3.3.5 Derivatives of the reflectance with respect to specific parameters

If polarization is ignored, the partial derivative of the reflectance with respect to some specific quantity $x(z)$ (e.g. ozone number density or aerosol number density) can be expressed as:

$$
\frac{\partial^2 R}{\partial x \partial z} = \frac{\partial^2 R}{\partial k_{\text{sca}} \partial z} \frac{\partial k_{\text{sca}}}{\partial x} + \frac{\partial^2 R}{\partial k_{\text{abs}} \partial z} \frac{\partial k_{\text{abs}}}{\partial x} + \sum_{l=0}^{\infty} \frac{\partial^2 R}{\partial \alpha_1^l \partial z} \frac{\partial \alpha_1^l}{\partial x}
\tag{27}
$$

where $R$ is the (1,1) element of $\mathbf{R}$ and $\alpha_1^l$ are the phase matrix expansion coefficients for unpolarized light. In DISAMAR the third term on the right-hand side is ignored, so it is assumed that the phase matrix of the particles does not change as function
of $z$. The derivatives for specific parameters can be calculated using these basic derivatives (De Haan, 2012). Here we give two examples of the use of the semi-analytical derivatives presented in the previous section.

The first example is the altitude-resolved air mass factor (also called block air mass factor) at wavelength $\lambda$, $m(z,\lambda)$. The altitude-resolved air mass factor is used in DOAS retrievals and represents the relative reduction in the reflectance when a unit amount of absorption is added to the atmosphere in a thin layer located between $z$ and $z + dz$. So it is the partial derivative of
the reflectance due to the absorber at $z$ divided by the reflectance. Writing the reflectance of the atmosphere without the extra absorption as $R(\lambda, 0)$, the altitude-resolved air mass factor is defined as:

$$
m(z,\lambda) = \frac{-1}{R(\lambda, 0)} \frac{\partial^2 R(\lambda, 0)}{\partial k_{\text{abs}}(z,\lambda) \partial z}.
\tag{28}
$$

The partial derivative on the right-hand side can now be calculated from Eq. 25.





The second example is the derivative of the reflectance to the altitude of a Lambertian cloud below an absorbing and scattering gas layer:

$$\frac{\partial R}{\partial z} = -\frac{\partial^2 R}{\partial k_{\text{gas,sca}}\partial z}(z)k_{\text{gas,sca}}(z) - \frac{\partial^2 R}{\partial k_{\text{gas,abs}}\partial z}(z)k_{\text{gas,abs}}(z). \tag{29}$$

Here the cloud is approximated as a Lambertian surface, without any transmission. When the altitude of the Lambertian cloud increases, the amount of gas above the cloud is reduced, so we get a minus sign in Eq. 29. The amount of scattering gas that is removed is proportional to the volume scattering coefficient ($k_{\text{gas,sca}}$) whereas the amount of absorbing gas that is removed is proportional to the volume absorption coefficient ($k_{\text{gas,abs}}$). The partial derivatives on the right-hand side of Eq. 29 can now be calculated from Eqs. 25 and 26.

As noted before, the derivatives can also be calculated numerically. Then $dR(\lambda, \mathbf{x})/dx_k$, with $x_k$ element $k$ of the state vector $\mathbf{x}$, is written as the numerical differential:

$$\frac{\partial R}{\partial x_k} = \frac{R(\lambda, \mathbf{x} + \Delta\mathbf{x}) - R(\lambda, \mathbf{x})}{\Delta x_k}. \tag{30}$$

Here $\Delta\mathbf{x} = 0$, except for the element $k$, where $\Delta\mathbf{x} = \Delta x_k$. By comparing the analytical derivatives in DISAMAR with the numerical derivatives, we can validate the former.

### 3.3.6 Layer-Based Orders of Scattering method (LABOS)

In the adding method as implemented in DISAMAR, the reflection matrix is calculated for a set of solar directions equal to the number of Gaussian division points used for integration over the polar angle plus at least two additional directions, namely the actual viewing direction and the actual solar position. Thus for satellite retrievals the adding method provides more results than required.

The Layer-Based Orders of Scattering method (LABOS) has been developed to obtain the reflection matrix for the actual solar position, or, when derivatives are required, for the actual solar position and a solar position corresponding to the viewing direction. The reflection and transmission properties of the individual homogeneous layers are still calculated with the doubling method; only the adding of different layers and the subsequent calculation of the internal field is replaced by the successive orders of scattering method. Here one order of scattering represents scattering by an atmospheric layer. This differs from the classical method of successive orders of scattering where the scattering element is a volume-element of the atmosphere instead of a layer (Lenoble et al., 2007; Min and Duan, 2004). In the adding method one deals with matrix-matrix multiplications, whereas in LABOS one deals with matrix-vector multiplications. However, in LABOS the calculations have to be repeated for the different orders of scattering. Hence, for an optically thin atmosphere with only a few orders of scattering LABOS requires less calculation time, while adding is more efficient for optically thick atmospheres, e.g. containing clouds.

In LABOS we deal with the upward and downward internal radiation fields, i.e. $\mathbf{U}$ and $\mathbf{D}$. In order to determine the derivatives using reciprocity, we need to consider two directions for the incident light, one corresponding to the actual solar position and one corresponding to the viewing direction. Therefore, $\mathbf{U}$ and $\mathbf{D}$ are $(4N, 2)$ matrices with the first column corresponding





to incident light for the viewing direction and the second column corresponding to the solar direction, where the number of
polar angles $N_\mu = N_g + 2$. If the incident light is polarized we need $(4N_\mu, 8)$ matrices.

Let $\mathbf{R}_k$ and $\mathbf{T}_k$ be the supermatrices for the reflection and transmission of layer $k$ ($k = 0, 1, ... K$). If $k = 0$, the reflection
matrix is that of the surface and the transmission matrix vanishes. Further, let $\mu$ and $\mu_0$ be the cosines of the viewing and solar
zenith angle, respectively. In order to evaluate different orders of scattering we first consider the so-called local radiation field
which is due to the reflection by the layer just below the level considered and the transmission by the layer just above this level.
Figure 5 illustrates the local upward and downward radiation fields at the interface $k$ for the first order scattering ($n = 1$).

The first order local upward radiation at level $k$ is given by:

$$\mathbf{U}_k^{\text{local},1} = \mathbf{R}_k \exp(-\frac{\tau_k}{\mu_0}) \tag{31}$$

where $\tau_k$ is the vertical optical depth of level $k$, measured from the top of the atmosphere. The two columns of matrix $\mathbf{U}_k^{\text{local},1}$
correspond to the two columns of supermatrix $\mathbf{R}_k$ having indices $4N_g + 1$ and $4(N_g + 1) + 1$, respectively, if the dimension of
the Stokes vector is 4. The first order local downward radiation field at level $k$ is:

$$\mathbf{D}_k^{\text{local},1} = \mathbf{T}_{k+1} \exp(-\frac{\tau_{k+1}}{\mu_0}). \tag{32}$$

Here the two columns of $\mathbf{D}_k^{\text{local},1}$ correspond to the two columns of supermatrix $\mathbf{T}_{k+1}$ with indices $4N_g + 1$ and $4N_g + 5$. The
superscript 1 denotes the first order of scattering for the layers.

From the local radiation fields we can now derive the total radiation fields. The first order scattered upward radiation at level
$k$, $\mathbf{U}_k^1$, is calculated as the sum of the contributions of the local upward radiation from the ground up to level $k$. The downward
radiation at level $k$, $\mathbf{D}_k^1$, is calculated as the sum of the contributions of the local downward radiation from level $k$ up to level
$K$ (TOA):

$$\mathbf{U}_k^1 = \sum_{l=0}^{k} \exp(-\frac{\tau_l - \tau_k}{\mu}) \mathbf{U}_l^{\text{local},1} \tag{33}$$

$$\mathbf{D}_k^1 = \sum_{l=k}^{K} \exp(-\frac{\tau_k - \tau_l}{\mu}) \mathbf{D}_l^{\text{local},1} \tag{34}$$

where the attenuation accounts for the direct transmission of light from level $l$ to level $k$. We note that Eqs. 33 and 34 can be
rewritten in terms of the following recurrence relations which are more efficient to evaluate:

$$\mathbf{U}_0^1 = \mathbf{U}_0^{\text{local},1} \tag{35}$$

$$\mathbf{U}_{k+1}^1 = \mathbf{U}_{k+1}^{\text{local},1} + \exp(-\frac{\tau_k - \tau_{k+1}}{\mu}) \mathbf{U}_k^1 \tag{36}$$

$$\mathbf{D}_K^1 = 0 \tag{37}$$

$$\mathbf{D}_{k-1}^1 = \mathbf{D}_{k-1}^{\text{local},1} + \exp(-\frac{\tau_{k-1} - \tau_k}{\mu}) \mathbf{D}_k^1. \tag{38}$$

The same recurrence relations hold for the higher orders of scattering, because the direct transmission of radiation to other
levels is the same for each order of scattering.





The following equations are used to determine the local radiation fields for the higher orders of scattering, where $n$ indicates
the order of scattering:

$$\mathbf{U}_k^{\mathrm{local},n+1} = \mathbf{R}_k \mathbf{D}_k^n + \mathbf{T}_k^* \mathbf{U}_{k-1}^n \tag{39}$$

$$\mathbf{D}_k^{\mathrm{local},n+1} = \mathbf{T}_{k+1} \mathbf{D}_{k+1}^n + \mathbf{R}_{k+1}^* \mathbf{U}_k^n \tag{40}$$

where $\mathbf{R}_{k+1}^*$ is the reflection of layer $k+1$ for illumination at its bottom. Since the layers are homogeneous we can use
symmetry relations to calculate $\mathbf{R}_{k+1}^*$ and $\mathbf{T}_k^*$ from $\mathbf{R}_{k+1}$ and $\mathbf{T}_k$, respectively (Hovenier et al., 2004). After the local radiation
fields are calculated using Eqs. 39 and 40, the recurrence relations are applied for the transfer to other levels to provide the
radiation fields $\mathbf{U}_k^{n+1}$ and $\mathbf{D}_k^{n+1}$ for scattering order $n+1$.

One can continue calculating the orders of scattering and summing them up, until the contribution of higher orders can
be ignored. At the end of the calculation the internal radiation fields at all levels are known. The Fourier coefficients of the
upward radiation at the top of the atmosphere provide the reflection, and the internal radiation fields are used to calculate the
380 derivatives. In DISAMAR users can choose to use either the doubling-adding method or the doubling-LABOS method.

### 3.3.7 Reflection calculated by integration over the source function

The upward radiation at the top of the atmosphere is calculated by assuming that the atmosphere is consisting of homogeneous
layers, both in the doubling-adding method and in the doubling-LABOS method. The calculation will converge to the correct
value if many homogeneous layers are used to represent the true atmospheric scattering and absorption profile. In DISAMAR
the reflection can also be calculated from numerical integration over the source function $\mathbf{J}$, to mitigate errors due to too coarse
layering. The reason is that by using the source function the assumption of homogeneous layers is not used for single scattering,
but only for multiple scattering. For each azimuthal Fourier term the reflection is then given by:

$$\mathbf{R}(\mu,\mu_0) = \exp(-\frac{\tau(0)}{\mu})\,\mathbf{U}(z=0,\mu,\mu_0) \;+\; \int_0^{\mathrm{TOA}} \exp(-\frac{\tau(z)}{\mu})\,\mathbf{J}(z,\mu,\mu_0)dz \tag{41}$$

where the surface is at $z=0$. Since the source function for the surface is a Dirac delta function in $z$, its contribution to the
390 reflection occurs as a separate term in Eq. 41. The source function for the atmosphere is:

$$\mathbf{J}(z,\mu,\mu_0) = \frac{1}{2\mu} \int_0^1 k_{\mathrm{sca}}(z)\,\mathbf{Z}(-\mu,-\mu')\,\mathbf{U}(z,\mu',\mu_0)d\mu'$$

$$+ \frac{1}{2\mu} \int_0^1 k_{\mathrm{sca}}(z)\,\mathbf{Z}(-\mu,\mu')\,\mathbf{D}(z,\mu',\mu_0)d\mu'$$

$$+ \frac{k_{\mathrm{sca}}}{4\mu\mu_0}\,\mathbf{Z}(-\mu,\mu_0)\exp(-\frac{\tau(z)}{\mu_0}). \tag{42}$$

$\mathbf{U}(z,\mu',\mu_0)$ and $\mathbf{D}(z,\mu',\mu_0)$ are the internal fields at altitude $z$ which are known after doubling-adding or doubling-LABOS
calculations. Hence, $\mathbf{J}$ can be easily calculated at those altitudes where the internal radiation fields are known.





## 4  DISAMAR retrieval methods

Three retrieval methods are implemented in DISAMAR: Optimal Estimation (OE), Differential Optical Absorption Spectroscopy (DOAS), and Differential and Smooth Absorption Separated (DISMAS).

The OE method is implemented in DISAMAR following Rodgers (2000). The OE method can be used for various retrievals implemented in DISAMAR, especially ozone profile retrieval (Kroon et al., 2011) and aerosol layer height retrieval (Sanders et al., 2015; Nanda et al., 2018). The OE output includes the retrieved parameters and their error matrices, correlations, gain matrices, and averaging kernels.

The DOAS method is described in the literature (Platt and Stutz, 2008). Below we describe the DISMAS method, which is related to the DOAS method, in detail.

### 4.1  DISMAS method

The DISMAS method uses the principle of DOAS for the forward simulation of spectra and the OE method for the retrieval. Therefore the DISMAS method reduces the forward simulation time, making it faster than a full forward simulation, but still provides the full error estimation as in OE. DISMAS can be used for the retrieval of total columns of weakly absorbing gases, like $O_3$, $NO_2$, $SO_2$, BrO, and $CH_2O$, but not for strong line absorbers. In DISMAS, gas absorption spectra are separated into a smooth part and a differential part, which is similar to DOAS. As an illustration, Fig. 6 shows an $NO_2$ absorption spectrum with a smooth and a differential absorption part.

For the reflectance spectrum around a weak atmospheric absorption line, e.g. from $NO_2$, one can use the Lambert-Beer attenuation law as follows:

$$R(\lambda, \sigma_a^{\mathrm{abs}}(\lambda)) = R(\lambda, 0) \exp(-M(\lambda) N \sigma_a^{\mathrm{abs}}(\lambda)) \tag{43}$$

where $R(\lambda, 0)$ is the reflectance of the atmosphere without the absorbing gas, $M(\lambda)$ is the air mass factor (AMF) of the total atmosphere (unitless), $\sigma_a^{\mathrm{abs}}(\lambda)$ is the altitude-averaged (subscript $a$) absorption cross-section (in $\mathrm{cm}^2/\mathrm{molecule}$), and $N$ is the vertical column density of the absorbing gas (in $\mathrm{molecule}/\mathrm{cm}^2$). Introducing the smooth absorption cross-section, $\sigma_a^{\mathrm{sm}}$, and the differential absorption cross-section, $\sigma_a^{\mathrm{dif}}$, one obtains:

$$R(\lambda, \sigma_a^{\mathrm{abs}}(\lambda)) = R(\lambda, \sigma_a^{\mathrm{sm}}(\lambda)) \exp(-M(\lambda) N \sigma_a^{\mathrm{dif}}(\lambda)). \tag{44}$$

Here $R(\lambda, \sigma_a^{\mathrm{sm}}(\lambda))$ and $M(\lambda)$ are smooth functions of wavelength, which are calculated with a radiative transfer model at a few wavelengths and are fitted as a function of wavelength to be interpolated to a high resolution wavelength grid. The differential term

$$\exp(-M(\lambda) N \sigma_a^{\mathrm{dif}}(\lambda)) \tag{45}$$

is varying strongly with wavelength and has to be evaluated at a high resolution wavelength grid. However, this analytical formula can be calculated very fast without the need to solve the radiative transfer model. The smooth functions $R(\lambda, \sigma_a^{\mathrm{sm}}(\lambda))$





and $M(\lambda)$ are calculated at those wavelengths, $\lambda_p$, where the differential absorption cross-sections are zero, because at those points (see Eq. 44):

$$R(\lambda, \sigma_a^{\text{abs}}(\lambda_p)) = R(\lambda, \sigma_a^{\text{sm}}(\lambda_p)). \tag{46}$$

Hence radiative transfer calculations for a small number of wavelengths $\lambda_p$ are sufficient to determine the reflectance spectrum at a high spectral resolution grid.

To simulate a measured spectrum in DISMAS, the modelled reflectance spectrum on the high resolution grid is multiplied with a high resolution solar irradiance spectrum, and convoluted with the instrument's spectral response function (or slit function). This provides a simulated radiance spectrum of the instrument that can be compared to an actually measured radiance spectrum. In practice, the simulated radiance spectrum is divided by the simulated irradiance spectrum to eliminate common errors, and the logarithm of the reflectance is used for fitting.

The air mass factor $M(\lambda)$ can be calculated from $m(z, \lambda)$, which is known from the radiative transfer calculations (see Eq. 28):

$$M(\lambda) = \frac{1}{N} \int\limits_0^{\text{TOA}} m(z, \lambda) n(z) dz, \tag{47}$$

where $n(z)$ is the number density profile of the absorbing gas. To start the optimal estimation process in DISMAS, $m(z, \lambda)$ is initialized using a geometrical air mass factor. In subsequent iteration steps the value of $m(z, \lambda)$ is updated by radiative transfer calculations for the new atmospheric composition state vector.

If there is more than one absorbing trace gas in a fit window one can use the slant absorption optical thickness, $\tau_{\text{slant}}^{\text{abs}}(\lambda)$, instead of the absorption cross-section to separate the smooth and differential parts of the absorption spectrum.

For fitting the modelled reflectance to the measured reflectance and for determining the errors in the retrieved parameter values, the derivatives are needed. Here we give the derivatives with respect to the total column of a trace gas and to the cloud fraction for a partly cloudy pixel.

For a partly cloudy pixel with $K$ different trace gases, the reflectance is written as:

$$R(\lambda) = (1-c) R_{\text{sm}}^{\text{clr}}(\lambda) \exp\left(-\sum_{k=1}^K M_k^{\text{clr}} N_k \sigma_{a,k}^{\text{dif}}(\lambda)\right) + c R_{\text{sm}}^{\text{cld}}(\lambda) \exp\left(-\sum_{k=1}^K M_k^{\text{cld}} N_k \sigma_{a,k}^{\text{dif}}(\lambda)\right) \tag{48}$$

where $c$ is the cloud fraction, subscript 'sm' refers to the smooth reflectance, and superscripts 'clr' and 'cld' denote the clear and cloudy part of the pixel, respectively. The derivative with respect to the cloud fraction is:

$$\frac{\partial R(\lambda)}{\partial c} = R^{\text{cld}}(\lambda) - R^{\text{clr}}(\lambda). \tag{49}$$





The derivative with respect to the total column of trace gas $k$, $N_k$, becomes:

$$
\begin{aligned}
\frac{\partial R(\lambda)}{\partial N_k} =\ & (1-c)\frac{\partial R_{\mathrm{sm}}^{\mathrm{clr}}(\lambda)}{\partial N_k}\exp\left(-\sum_{j=1}^{K}M_j^{\mathrm{clr}}\,N_j\,\sigma_{a,j}^{\mathrm{dif}}(\lambda)\right) \\
& -(1-c)R^{\mathrm{clr}}(\lambda)\,M_k^{\mathrm{clr}}(\lambda)\,\sigma_{a,k}^{\mathrm{dif}}(\lambda) \\
& +c\frac{\partial R_{\mathrm{sm}}^{\mathrm{cld}}(\lambda)}{\partial N_k}\exp\left(-\sum_{j=1}^{K}M_j^{\mathrm{cld}}\,N_j\,\sigma_{a,j}^{\mathrm{dif}}(\lambda)\right) \\
& -cR^{\mathrm{cld}}(\lambda)\,M_k^{\mathrm{cld}}(\lambda)\,\sigma_{a,k}^{\mathrm{dif}}(\lambda)
\end{aligned}
\tag{50}
$$

where we assume that the air mass factor does not change significantly when the column changes, which is accurate for weak
absorption.

## 5 Examples of applications

### 5.1 Comparison with DAK

As an example, we show a comparison of DISAMAR with the Doubling-Adding KNMI (DAK) radiative transfer model (De
Haan et al., 1987; Stammes et al., 1989) for the simulated reflectance and linear polarization spectrum of a Rayleigh scattering
atmosphere in the ozone Huggins bands region between 325 and 335 nm; see Fig. 7. In DISAMAR we used the doubling-adding
method in order to get the best agreement with DAK. The atmospheric temperature, pressure and ozone mixing ratio profiles
are taken from the mid-latitude summer atmosphere (Anderson et al., 1986). Since DISAMAR uses the hydrostatic relation to
calculate altitude with the input pressure and temperature profiles, the altitude in DAK is corrected for the hydrostatic relation.
This correction is important to obtain the same Rayleigh scattering optical thickness in this wavelength range. The total ozone
column is 335.66 DU in DAK and DISAMAR. The ozone cross section was taken from Bass and Paur (1985), because this
data is available in both DAK and DISAMAR.

Figure 7 shows the simulated reflectance, degree of linear polarization and the corresponding relative differences, for the
viewing and solar geometry $\mu = 1$, $\mu_0 = 0.5$. DISAMAR and DAK agree very well for the simulated reflectance and degree
of linear polarization, with absolute differences less than 0.01%. DISAMAR and DAK use different internal wavelength grids,
altitude grids and interpolation methods, which may explain the remaining small differences.

### 5.2 Derivatives

The accuracy of the semi-analytical formulas for the derivatives based on the adding method is checked here by comparison
with the numerically calculated derivatives. We show an example of the derivative of the cloud height, $dR/dz$, in the $O_2$
A-band for a Lambertian cloud. In the calculation, the atmospheric profile is mid-latitude summer, the cloud fraction is set
to 1 and the cloud albedo to 0.8. The viewing and solar geometry is $\mu = 1$, $\mu_0 = 0.8660$. The spectrum of the derivative is
calculated at a high resolution wavelength grid and convolved with a Gaussian ISRF with a FWHM of 0.4 nm. As shown in





Fig. 8 the semi-analytical and numerical derivatives are very similar: the mean relative difference is -0.13%. At 758–759 nm, the $O_2$ absorption is very small, about 100 times smaller than in the absorption band close to 760–761 nm. The spectrum of $dR/dz$ clearly shows that the cloud height information comes from the absorption band. If the cloud fraction is smaller than 1,

the derivative is also smaller, because the radiance is smaller. Although the relative difference between the semi-analytical and numerical derivatives behaves steeply in the continuum at 758–759 nm, it is still within 0.2% over the entire absorption band. Therefore the semi-analytical derivatives are considered accurate for cloud height retrievals.

The altitude-resolved AMF, $m(\lambda, z)$, is used in the DOAS algorithm to convert the slant column density of a gas to the vertical column density (see Sect. 3.3.5). The altitude-resolved AMF was calculated using Eq. 28 and the semi-analytical

partial derivatives from DISAMAR. It can also be calculated numerically, but that is slower and requires a finer altitude grid to get accurate results. Figure 9 shows the altitude-resolved AMF of $NO_2$ calculated using the semi-analytical derivatives for a cloudy scene and a clear-sky scene. Here the atmospheric profile and $NO_2$ mixing ratio are taken from the mid-latitude summer atmospheric profile. A scattering cloud layer is specified between 800 and 700 hPa with a Henyey-Greenstein phase function, asymmetry parameter 0.85, and single scattering albedo 1.0. The cloud optical thickness is 10. The surface albedo is 0.05. The

viewing and solar geometry is $\mu = 1$, $\mu_0 = 0.8660$.

Figure 9 shows that inside the cloud the AMF has a peak close to the cloud top due to the enhancement from multiple scattering. Above the cloud top, the AMF is larger than the clear-sky AMF due to the bright cloud layer. Below the cloud top, the AMF decreases and becomes smaller than the clear-sky AMF. At TOA, the AMF for both scenes approach the geometric AMF, $1/\mu + 1/\mu_0$, which is 2.155 for this case. These are typical characteristics of the altitude-resolved AMF of $NO_2$.

## 5.3 Comparison of DISMAS and full OE for $NO_2$ retrievals

The DISAMAR code can simulate the retrieval of total columns of trace gases, ozone profile, cloud layer height, aerosol layer height, cloud (or aerosol) optical thickness, surface albedo and surface pressure. Some OE retrieval applications were mentioned in the introduction of Sect. 4. Because the DISMAS retrieval approach is new and not used operationally, we show here as an example the DISMAS results for $NO_2$ column retrieval in a cloudy scene. The scene in the simulation is constructed

as follows: the atmosphere is a typical European-polluted model, the cloud is a Lambertian cloud with a fixed albedo of 0.8, located at 4 km altitude, and the geometry is nadir view with solar zenith angle 60°. In the retrieval the fitting window is 420–450 nm. Radiative transfer calculations are performed at 4 wavelengths for DISMAS and at about 900 wavelengths for OE. So DISMAS enables a large reduction of the number of radiative transfer calculations needed for spectral fitting. Table 1 shows the comparison between DISMAS and full OE for the $NO_2$ retrievals for two cases. In Case 1 three parameters are

retrieved: $NO_2$ total column density, surface albedo, and cloud fraction. In Case 2, the cloud fraction is fixed, and only the $NO_2$ total column density and the surface albedo are retrieved.

The parameters were accurately retrieved by both methods. The error estimates for the retrieved surface albedo and cloud fraction differ only little, but DISMAS underestimates the error in the retrieved $NO_2$ column. If the cloud fraction is fixed to its true value (0.20), so the $NO_2$ column and the surface albedo are the only fit parameters, then DISMAS and OE fully agree

for the errors in the $NO_2$ column, while the surface albedo values are also the same.



From our results we find that DISMAS and OE are functionally equivalent for weak absorption, except that the error in the retrieved column may differ in some cases. The main advantage is that DISMAS is much faster than full OE, which might make operational use of DISMAS attractive. This would make it possible to derive the surface albedo and the cloud fraction directly from the measured radiance in the $NO_2$ window, thereby reducing the bias of the retrieved $NO_2$ column significantly.

## 6 Summary and concluding remarks

The aim of the DISAMAR code is to determine instrument specifications and analyze methods for retrievals of atmospheric composition from satellite observations in the range of 270–2400 nm. We have described the principles of the DISAMAR radiative transfer model and its innovative retrieval capabilities. The novel features in DISAMAR are the semi-analytical derivatives of the reflectance using the internal radiation field, the layer-based orders of scattering method, LABOS, which speeds up the calculations, and the DISMAS retrieval method which reduces the number of radiative transfer calculations in DOAS-type retrievals. The semi-analytical derivatives are based on the linearization of the adding method, which is detailed in Appendix A. Other features of DISAMAR that improve the accuracy of forward modelling are the Gaussian grids for height and wavelength, and the integration over the source function to obtain the reflectance.

DISAMAR can be used as an accurate radiative transfer model for simulations and as a retrieval algorithm for several applications. We have given a few examples of applications, namely forward modelling of the polarized reflectance in the UV, the derivative of cloud height in the $O_2$ A-band, and an application of DISMAS for $NO_2$ retrieval in a cloudy scene. It is shown that for retrieval of total columns of weak absorbers, DISMAS has a similar functionality as Optimal Estimation but it is much faster.

In the past years DISAMAR has been used extensively in trace gas, aerosol and cloud retrieval studies for OMI, TROPOMI, and Sentinel-4/5. DISAMAR is currently being used for operational retrievals of ozone profiles from UV spectra for OMI (Kroon et al., 2011) and TROPOMI (Veefkind et al., 2021), and aerosol layer height retrieval from the $O_2$ A-band for TROPOMI (Nanda et al., 2019). More applications of DISAMAR are described in the DISAMAR algorithm document (De Haan, 2012), and can be found in the refereed literature; for example, on the role of aerosols in retrievals of $NO_2$, see Castellanos et al. (2015) and Chimot et al. (2016, 2017).

Currently, DISAMAR is still under development and has certain limitations: Raman scattering in water (Vasilkov et al., 2002) is ignored; remote sensing from the ground and from aircraft is not supported; and limb observations are not supported. Furthermore, the radiative transfer part assumes a plane-parallel atmosphere, albeit with a correction for the curvature of the atmosphere for incident sunlight (pseudo-spherical correction, see Appendix B). This may become inaccurate for observation far from nadir (viewing zenith angle > 60°), in particular for stratospheric gases (Spurr, 2002). In the coming years DISAMAR will be further extended and used to develop optimized retrieval algorithms, including those using neural networks (Nanda et al., 2019).



*Code and data availability.* The Disamar code v4.1.5 and data are available at Zenodo DOI 10.5281/zenodo.6304984,
https://zenodo.org/record/6304984#.Yjb_uBso_0o.
The Disamar code is also available at Gitlab https://gitlab.com/KNMI-OSS/disamar.

*Author contributions.* JdH developed the DISAMAR codes and wrote the algorithm document and manual. PW wrote the paper based on
the algorithm document, performed the calculations and made the figures. MS and JPV contributed to the DISAMAR codes. MS and PW
maintain the codes. PS contributed to the writing of the paper. All co-authors commented on the paper.

*Competing interests.* No competing interests.

*Acknowledgements.* Over the years many people have contributed to the testing of the code and developing applications. We want to thank
Mark ter Linden (Science and Technology B.V., Delft, the Netherlands) for building the software interface around DISAMAR.

*Financial support.*

The development of DISAMAR was funded by the Netherlands Space Office (NSO), under the OMI and TROPOMI science
contracts, and by ESA under various future mission contracts.





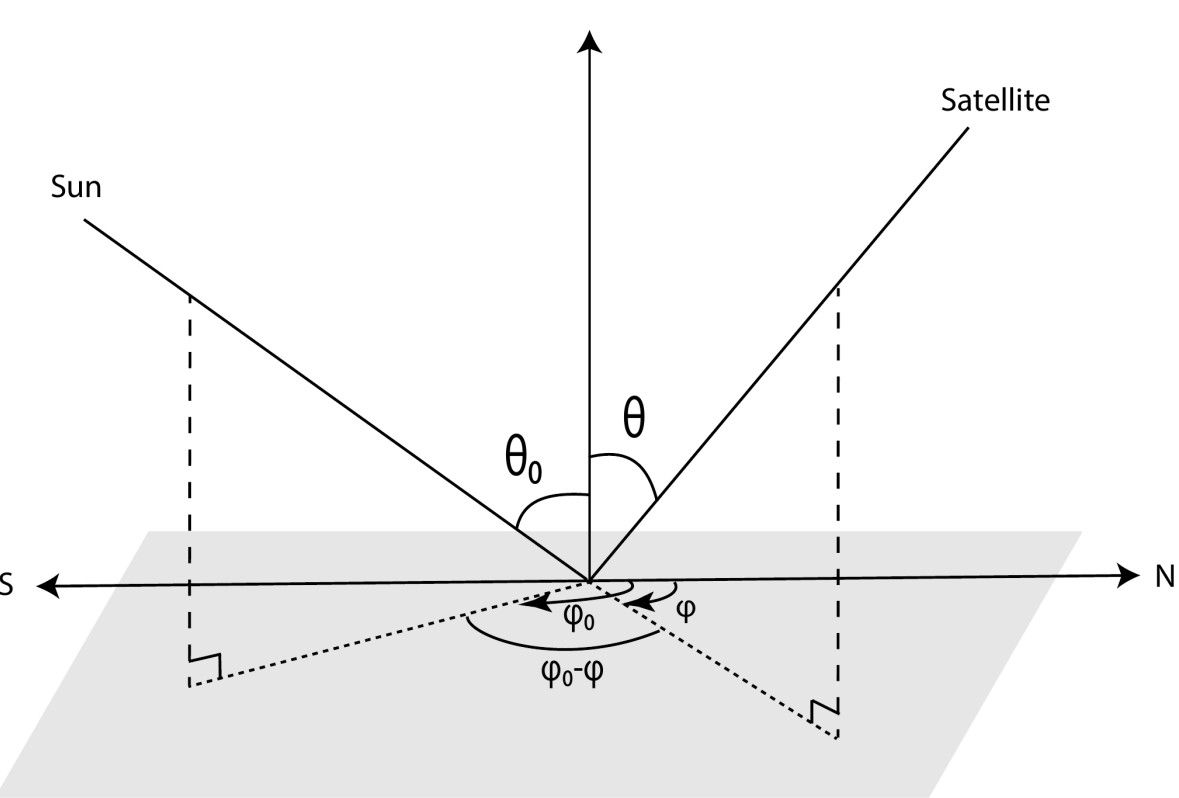

**Figure 1.** Definition of directions in DISAMAR. $\theta_0$ and $\phi_0$ are the solar zenith angle and the solar azimuth angle, respectively, and $\theta$ and $\phi$ are the satellite viewing zenith angle and satellite azimuth angle, respectively. The relative azimuth angle is $\phi - \phi_0$. In the supermatrices we use $\mu_0 = \cos\theta_0$ and $\mu = \cos\theta$.



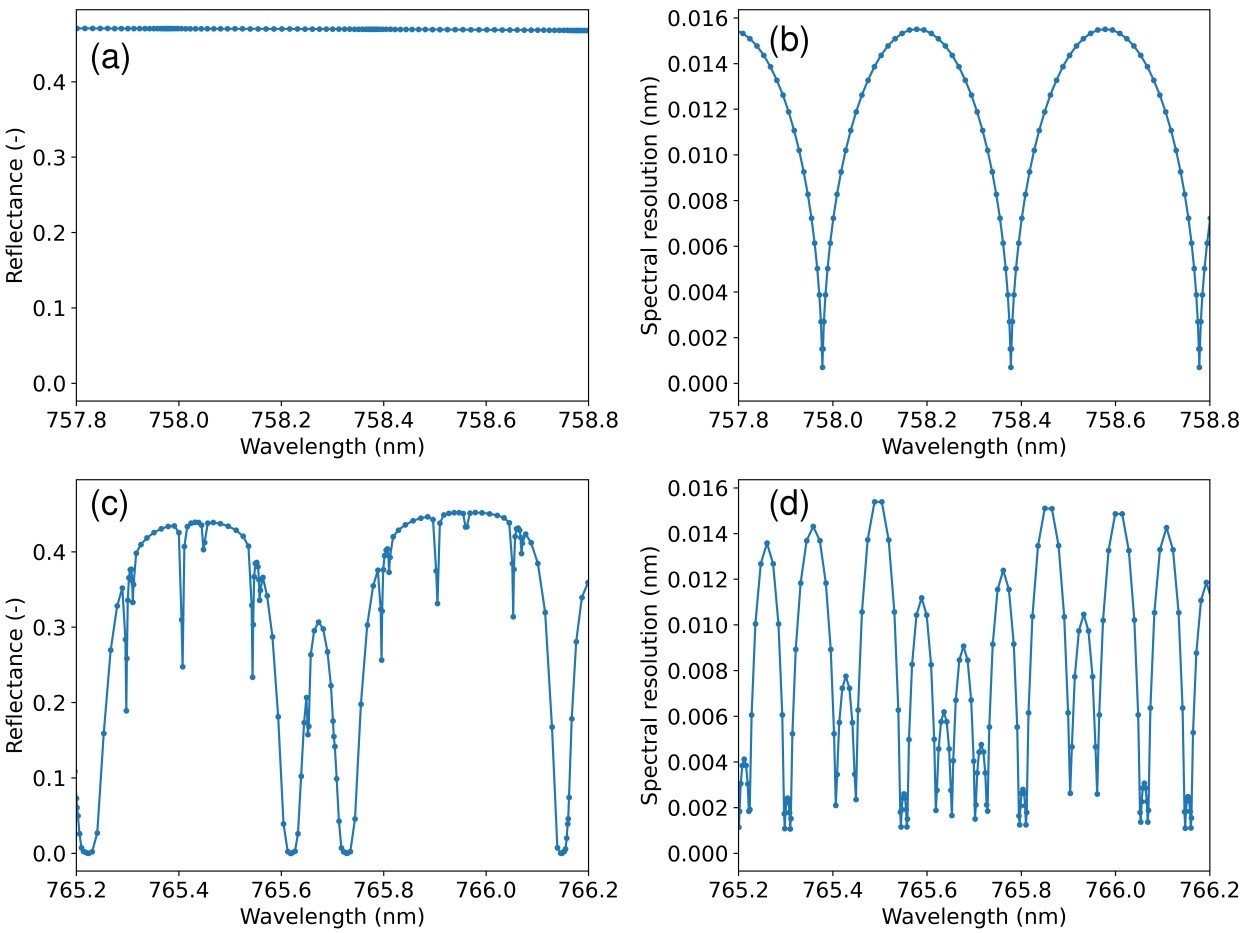

**Figure 2.** Illustration of the wavelength grid used in DISAMAR: (a) Simulated reflectance spectrum at top-of-atmosphere in the continuum part of the oxygen A-band, from 757.8-758.8 nm, at a Gaussian wavelength grid. (b) Wavelength grid interval distribution for the spectrum of (a). (c, d) are similar to (a,b) but for the oxygen A-band absorbing part 765.2-766.2 nm. In determining the Gaussian wavelength grid the positions of the absorption lines are taken into account. The number of Gaussian point is 40 and minimum 8.



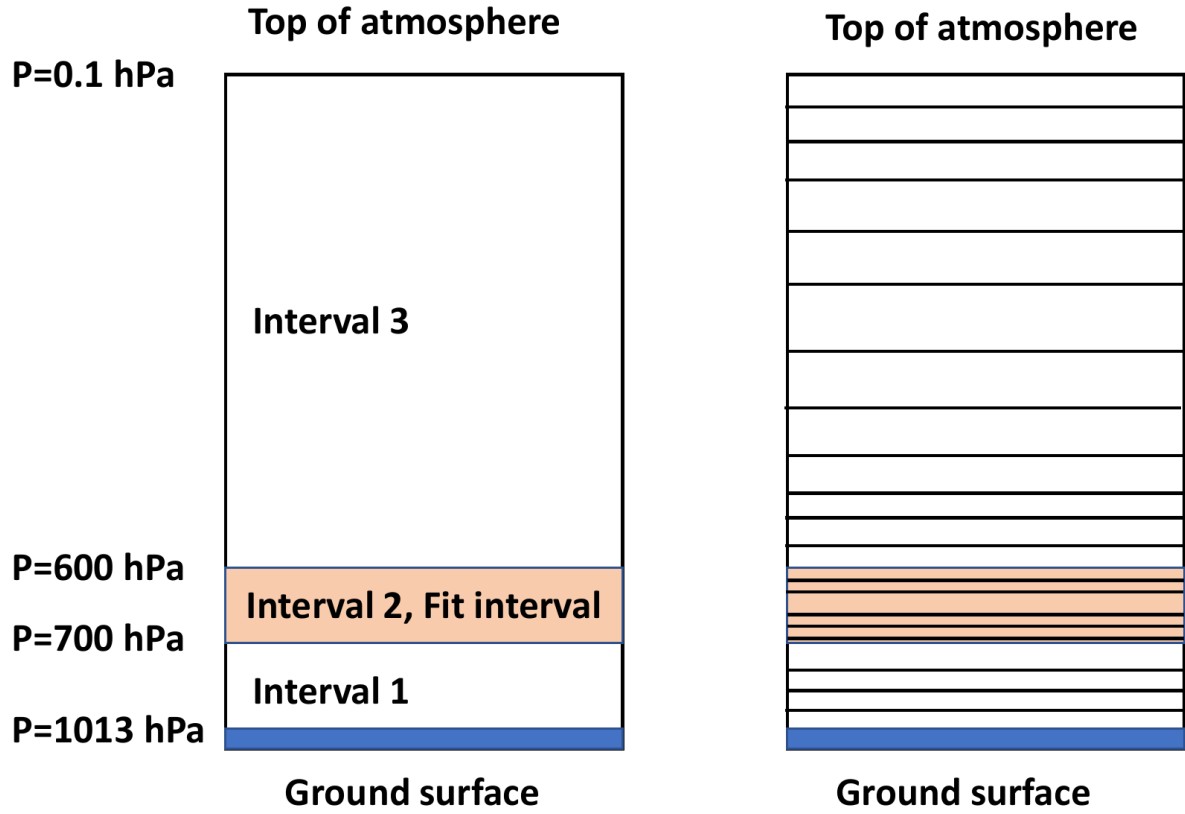

**Figure 3.** Schematic illustration of the altitude grid used in DISAMAR. Left: Pressure intervals in the atmosphere. Right: Each pressure interval is subdivided into layers using Gaussian division points in altitude, which are different per interval. Lowest level is the ground surface, highest level is the top-of-atmosphere.



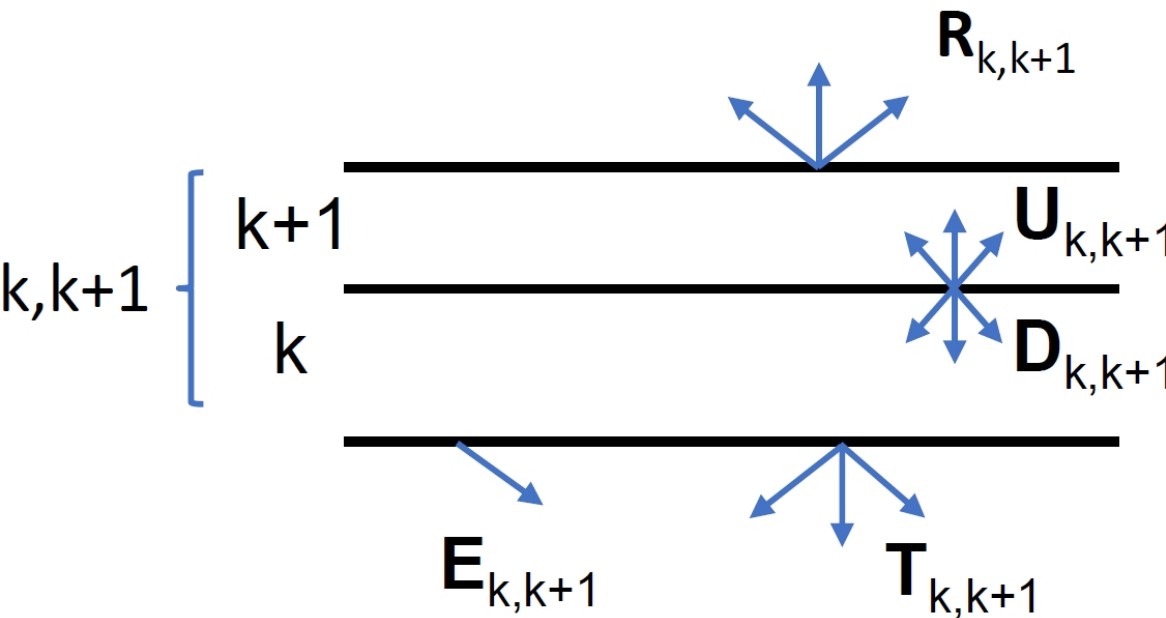

**Figure 4.** Radiation fields after adding two layers, $k$ and $k+1$, in the adding method. Layer $k, k+1$ is the combined layer. $\mathbf{U}_{k,k+1}$ and $\mathbf{D}_{k,k+1}$ are the upward and downward diffuse internal fields for the combined layer. $\mathbf{R}_{k,k+1}$ and $\mathbf{T}_{k,k+1}$ are the reflection and diffuse transmission of the combined layer. $\mathbf{E}_{k,k+1}$ is the combined direct transmission (direct attenuation).



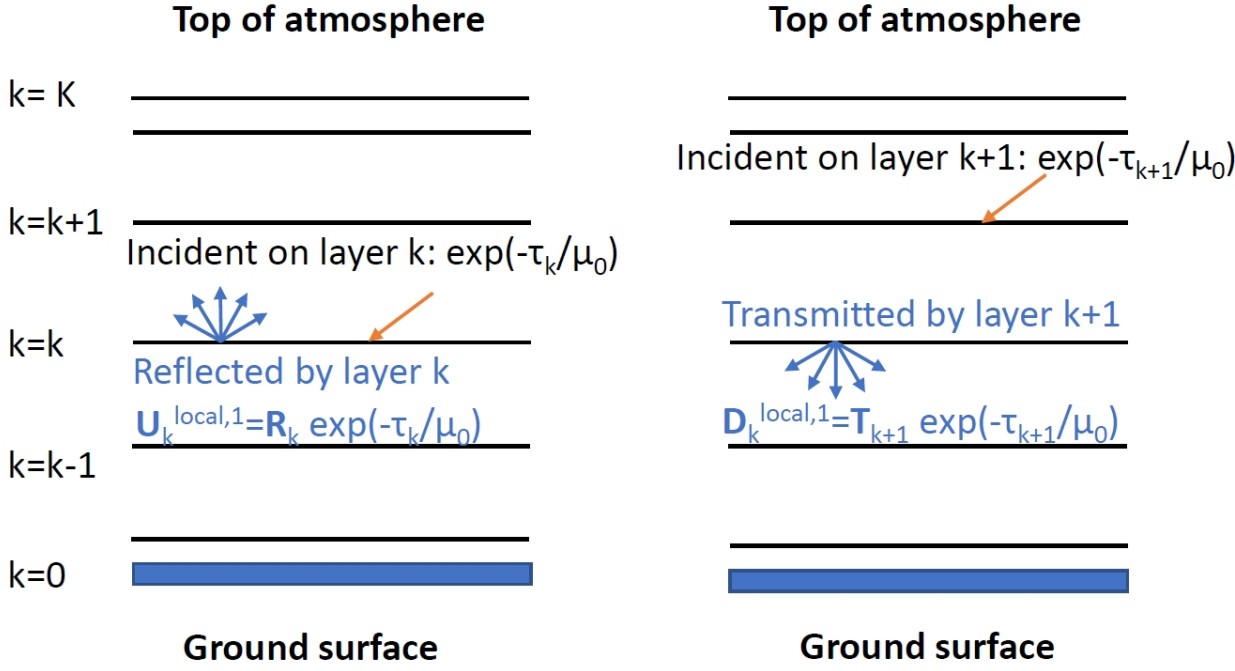

**Figure 5.** Illustration of the principle of the LABOS method, showing the first order scattering local upward ($\mathbf{U}_k^{\mathrm{local},1}$) and downward ($\mathbf{D}_k^{\mathrm{local},1}$) radiation fields at level $k$. The index $k = 0, .., K$ indicates the level, where $k = 0$ is the surface and $k = K$ is the top-of-atmosphere. The layer number refers to the upper level of the layer, so layer $k$ is located between levels $k - 1$ and $k$.

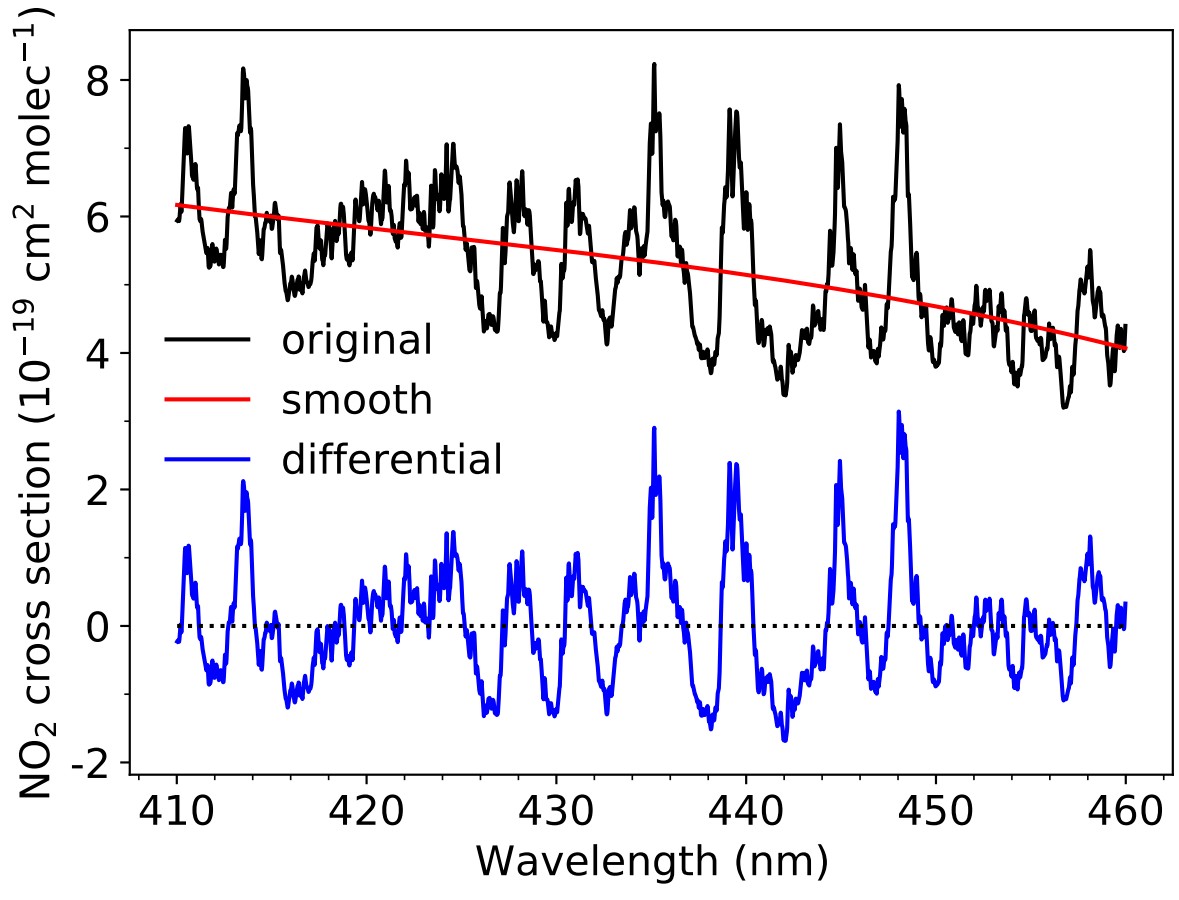

**Figure 6.** Illustration of the principle of DISMAS using the $NO_2$ cross-section as an example. The $NO_2$ cross-section is separated in two parts: a smooth part, using a fit of a third-order polynomial, and a differential part. The radiative transfer calculations are performed for the smooth part at a few selected wavelengths where the differential cross-section is vanishing.



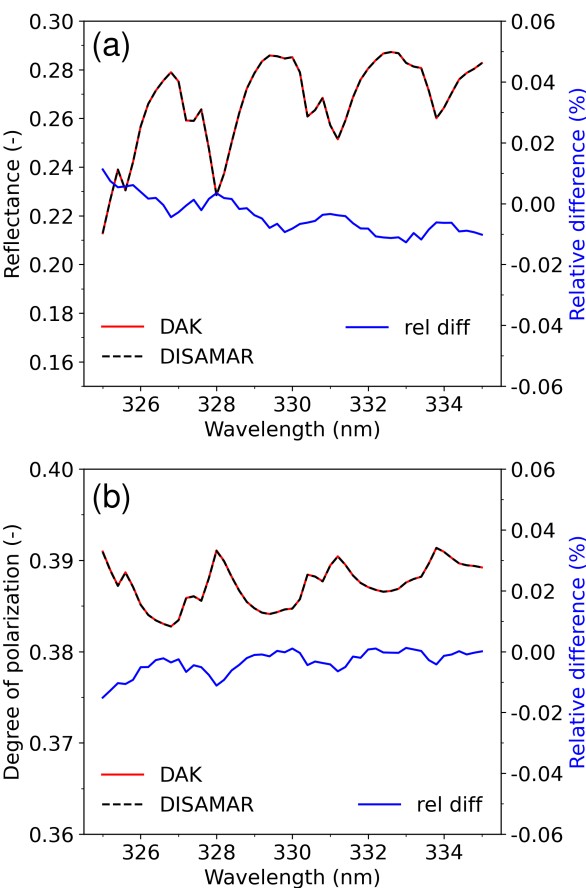

**Figure 7.** Comparison of DISAMAR and DAK simulations of TOA reflection spectra between 325 and 335 nm: (a) reflectance, and (b) degree of linear polarization. The relative difference is defined as (DISAMAR-DAK)/DAK (in percent). The wavelength step in the spectra is 0.2 nm and no ISRF convolution is performed. Geometry: nadir view, solar zenith angle 60°. Surface albedo is 0.02.





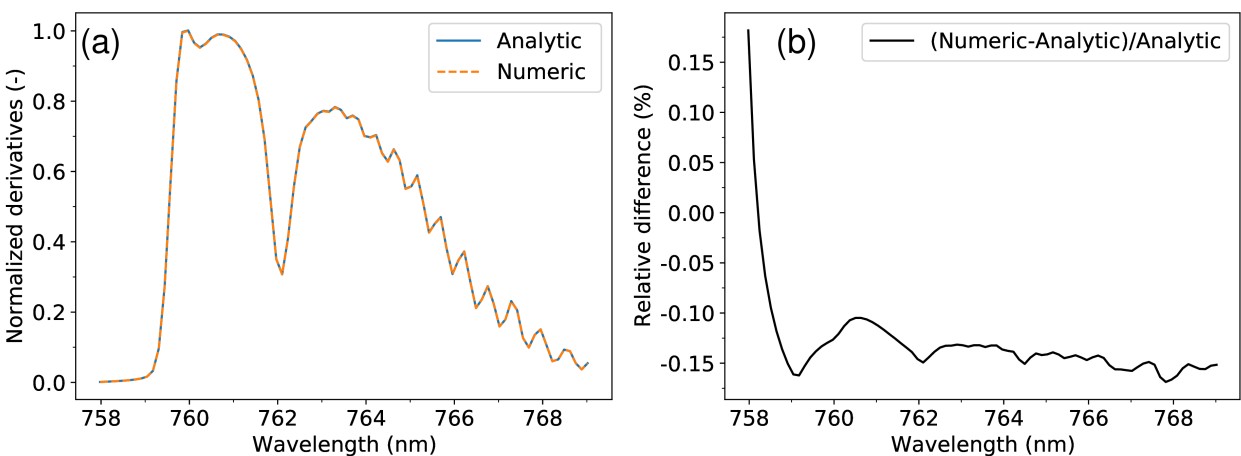

**Figure 8.** (a) Derivative of the reflectance to the cloud height, $dR/dz$, for a scene with a Lambertian cloud, calculated using the semi-analytical method (Eq. 29) and the numerical method. The derivative is normalized to the maximum value calculated using the numerical method. (b) Relative difference (in percent) between semi-analytical and numerical derivatives. The steep change in the relative difference at 758 nm is caused by the fact that the derivative is very small where the $O_2$ absorption approaches zero. Geometry: nadir view, solar zenith angle 30°. Cloud fraction is 1 and Lambertian cloud albedo is 0.8.

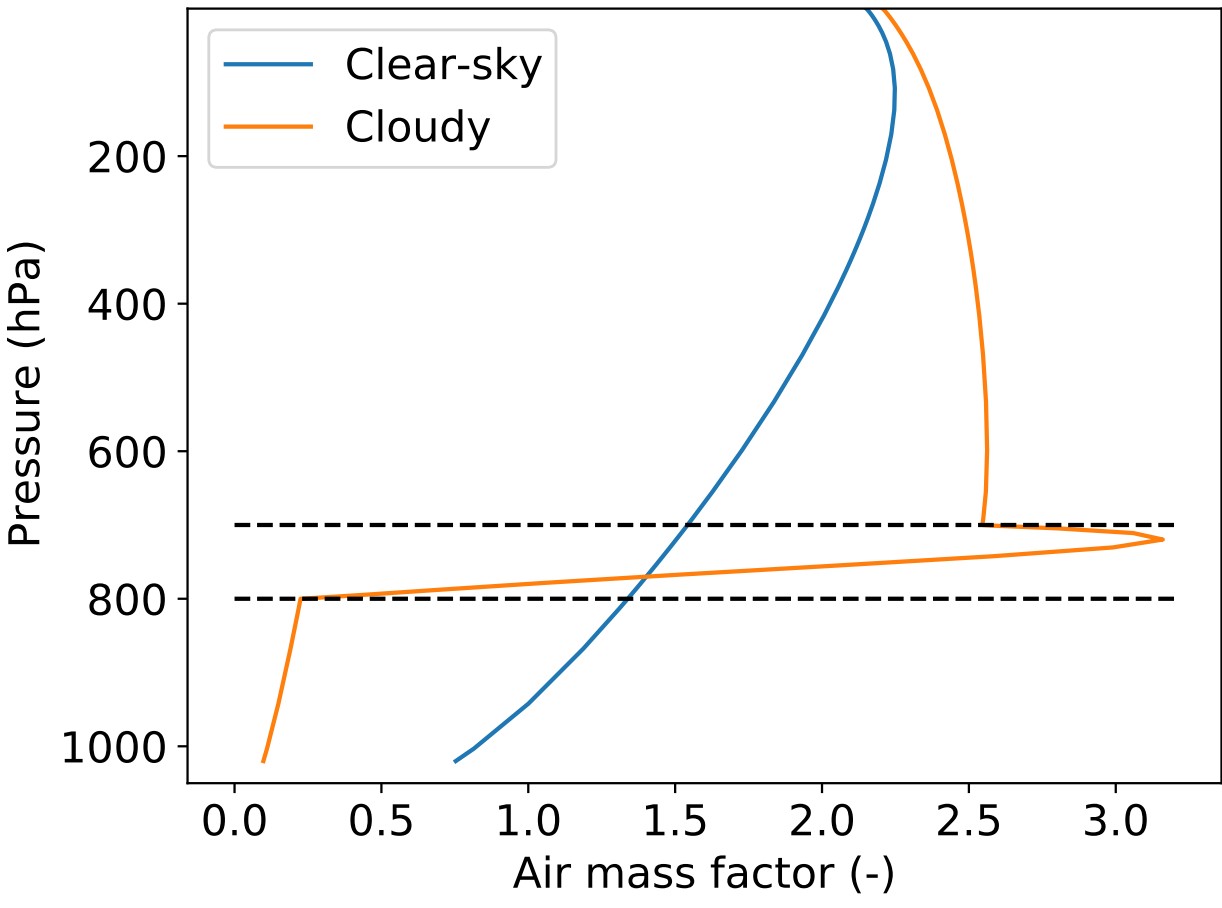

**Figure 9.** Altitude-resolved air mass factor of $NO_2$ at 440 nm for a cloudy scene and a clear-sky scene. The scattering cloud layer is located between 700 and 800 hPa with a cloud optical thickness of 10. The dashed lines indicate the cloud top and base pressure. The surface albedo is 0.05. Geometry: nadir view, solar zenith angle 30°.





**Table 1.** Comparison between DISMAS and Full Optimal Estimation (OE) retrievals of the total $NO_2$ column in a partly cloudy scene (cloud fraction 0.2, Lambertian cloud albedo 0.8) with a reflecting surface. In Case 1 all three parameters are retrieved, whereas in Case 2 the cloud fraction is fixed and only $NO_2$ column and surface albedo are retrieved. The a priori and retrieval errors are given between brackets.

| Case | Parameter | A priori | True | Full OE | DISMAS |
|------|-----------|----------|------|---------|--------|
| 1 | $NO_2$ ($\times 10^{16}$ molec cm$^{-2}$) | 1.0 (2.0) | 2.231 (-) | 2.228 (0.0952) | 2.245 (0.0671) |
| | Surface albedo | 0.03 (0.316) | 0.05 (-) | 0.0501 (0.00290) | 0.0498 (0.00223) |
| | Cloud fraction | 0.2 (1.0) | 0.2 (-) | 0.1999 (0.0027) | 0.2002 (0.0020) |
| 2 | $NO_2$ ($\times 10^{16}$ molec cm$^{-2}$) | 1.0 (2.0) | 2.231 (-) | 2.230 (0.0559) | 2.232 (0.0561) |
| | Surface albedo | 0.03 (0.316) | 0.05 (-) | 0.0500 (0.000130) | 0.0500 (0.000130) |





**Appendix A: Derivation of the derivative formulas by means of linearization**

Here we derive the formulas for the derivative of the reflection, $d\mathbf{R}/dz$, given by Eq. 15 in Sect. 3.3.4, by means of linearization of the adding equations. Thereto we divide the atmosphere into two parts, a top part and a bottom part. Let $\mathbf{R}_{\mathrm{top}}$, $\mathbf{R}_{\mathrm{top}}^*$, $\mathbf{T}_{\mathrm{top}}$, and $\mathbf{T}_{\mathrm{top}}^*$ be the Fourier coefficients of the reflection and transmission matrices of the top part. The quantities without an asterisk denote the properties for illumination at its top, while the asterisk denotes illumination at its bottom. Analogously, matrices with the subscript 'bot' denote the properties of the lower partial atmosphere. We start with the adding equations Eqs. 11, 10,

and 9, and change subscript $k+1$ to 'top' and subscript $k$ to 'bot':

$$\mathbf{R}_{\mathrm{top+bot}} = \mathbf{R}_{\mathrm{top}} + \mathbf{E}_{\mathrm{top}}\mathbf{U}_{\mathrm{top+bot}} + \mathbf{T}_{\mathrm{top}}^*\mathbf{U}_{\mathrm{top+bot}} \tag{A1}$$

$$\mathbf{U}_{\mathrm{top+bot}} = \mathbf{R}_{\mathrm{bot}}\mathbf{E}_{\mathrm{top}} + \mathbf{R}_{\mathrm{bot}}\mathbf{D}_{\mathrm{top+bot}} \tag{A2}$$

$$\mathbf{D}_{\mathrm{top+bot}} = \mathbf{T}_{\mathrm{top}} + \mathbf{Q}_{\mathrm{top+bot}}\mathbf{E}_{\mathrm{top}} + \mathbf{Q}_{\mathrm{top+bot}}\mathbf{T}_{\mathrm{top}}. \tag{A3}$$

Inserting Eqs. A2 and A3 into Eq. A1, we obtain the adding equation for the reflection at the top in a concise form:

$$\mathbf{R}_{\mathrm{top+bot}} = \mathbf{R}_{\mathrm{top}} + (\mathbf{E}_{\mathrm{top}} + \mathbf{T}_{\mathrm{top}}^*)\mathbf{R}_{\mathrm{bot}}(1 + \mathbf{Q}_{\mathrm{top+bot}})(\mathbf{E}_{\mathrm{top}} + \mathbf{T}_{\mathrm{top}}) \tag{A4}$$

where

$$\mathbf{Q}_{\mathrm{top+bot}} = \mathbf{R}_{\mathrm{top}}^*\mathbf{R}_{\mathrm{bot}} + \mathbf{R}_{\mathrm{top}}^*\mathbf{R}_{\mathrm{bot}}\mathbf{R}_{\mathrm{top}}^*\mathbf{R}_{\mathrm{bot}} + .... \tag{A5}$$

We now start the linearization process by adding a thin layer on top of the lower partial atmosphere. Using Eq. A4 and writing 'thin' instead of 'top' yields:

$$\mathbf{R}_{\mathrm{thin+bot}} = \mathbf{R}_{\mathrm{thin}} + (\mathbf{E}_{\mathrm{thin}} + \mathbf{T}_{\mathrm{thin}}^*)\mathbf{R}_{\mathrm{bot}}(1 + \mathbf{Q}_{\mathrm{thin+bot}})(\mathbf{E}_{\mathrm{thin}} + \mathbf{T}_{\mathrm{thin}}) \tag{A6}$$

$$\mathbf{Q}_{\mathrm{thin+bot}} = \mathbf{R}_{\mathrm{thin}}^*\mathbf{R}_{\mathrm{bot}} + \mathbf{R}_{\mathrm{thin}}^*\mathbf{R}_{\mathrm{bot}}\mathbf{R}_{\mathrm{thin}}^*\mathbf{R}_{\mathrm{bot}} + .... \tag{A7}$$

The radiation fields for a thin layer have already been given by Eqs. 1–5 in Sect. 3.3.2. We observe that $\mathbf{R}_{\mathrm{thin}}$, $\mathbf{T}_{\mathrm{thin}}$, $\mathbf{R}_{\mathrm{thin}}^*$, $\mathbf{T}_{\mathrm{thin}}^*$, as well as $(1 - \mathbf{E}_{\mathrm{thin}})$, are proportional to $dz$. In the linearization process we keep only terms that are constant or linear in $dz$. We note that only the first term of Eq. A7 has to be kept, because the second term is $\propto (dz)^2$. Inserting Eq. A7 into Eq.

A6 and keeping only terms that are constant or linear in $dz$, the derivation steps towards the linearized form of Eq. A6, given by Eq. A8, are:

$$\begin{aligned}
\mathbf{R}_{\mathrm{thin+bot}} &= \mathbf{R}_{\mathrm{thin}} + (\mathbf{E}_{\mathrm{thin}} + \mathbf{T}_{\mathrm{thin}}^*)\mathbf{R}_{\mathrm{bot}}(1 + \mathbf{Q}_{\mathrm{bot,thin}})(\mathbf{E}_{\mathrm{thin}} + \mathbf{T}_{\mathrm{thin}}) \\
&= \mathbf{R}_{\mathrm{thin}} + (\mathbf{E}_{\mathrm{thin}} + \mathbf{T}_{\mathrm{thin}}^*)\mathbf{R}_{\mathrm{bot}}(1 + \mathbf{R}_{\mathrm{thin}}^*\mathbf{R}_{\mathrm{bot}})(\mathbf{E}_{\mathrm{thin}} + \mathbf{T}_{\mathrm{thin}}) \\
&= \mathbf{R}_{\mathrm{thin}} + [1 + \mathbf{T}_{\mathrm{thin}}^* - (1 - \mathbf{E}_{\mathrm{thin}})]\mathbf{R}_{\mathrm{bot}}(1 + \mathbf{R}_{\mathrm{thin}}^*\mathbf{R}_{\mathrm{bot}})[1 + \mathbf{T}_{\mathrm{thin}} - (1 - \mathbf{E}_{\mathrm{thin}})] \\
&= \mathbf{R}_{\mathrm{thin}} + \mathbf{R}_{\mathrm{bot}} + [\mathbf{T}_{\mathrm{thin}}^* - (1 - \mathbf{E}_{\mathrm{thin}})]\mathbf{R}_{\mathrm{bot}} + \mathbf{R}_{\mathrm{bot}}[\mathbf{T}_{\mathrm{thin}} - (1 - \mathbf{E}_{\mathrm{thin}})] + \mathbf{R}_{\mathrm{bot}}\mathbf{T}_{\mathrm{thin}}^*\mathbf{R}_{\mathrm{bot}}.
\end{aligned} \tag{A8}$$





We now add the upper partial atmosphere to this combined ('thin+bot') lower layer, and derive the linearized expression for

$\mathbf{R}_{\text{top+thin+bot}}$, given by Eq. A16. We start again from Eq. A4:

$$\mathbf{R}_{\text{top+thin+bot}} = \mathbf{R}_{\text{top}} + (\mathbf{E}_{\text{top}} + \mathbf{T}^*_{\text{top}})\mathbf{R}_{\text{thin+bot}}(1 + \mathbf{Q}_{\text{top+thin+bot}})(\mathbf{E}_{\text{top}} + \mathbf{T}_{\text{top}}) \tag{A9}$$

$$\mathbf{R}_{\text{top+bot}} = \mathbf{R}_{\text{top}} + (\mathbf{E}_{\text{top}} + \mathbf{T}^*_{\text{top}})\mathbf{R}_{\text{bot}}(1 + \mathbf{Q}_{\text{top+bot}})(\mathbf{E}_{\text{top}} + \mathbf{T}_{\text{top}}). \tag{A10}$$

Subtracting these two equations we get the difference in $\mathbf{R}$ due to the thin layer:

$$\mathbf{R}_{\text{top+thin+bot}} - \mathbf{R}_{\text{top+bot}} = (\mathbf{E}_{\text{top}} + \mathbf{T}^*_{\text{top}})\left[\mathbf{R}_{\text{thin+bot}}(1 + \mathbf{Q}_{\text{top+thin+bot}}) - \mathbf{R}_{\text{bot}}(1 + \mathbf{Q}_{\text{top+bot}})\right](\mathbf{E}_{\text{top}} + \mathbf{T}_{\text{top}}). \tag{A11}$$

Let us now focus on the term between right brackets on the right-hand side of Eq. A11. The linearization steps, in which only constant terms and terms linear in $dz$ are kept, are:

$$\mathbf{R}_{\text{thin+bot}}(1 + \mathbf{Q}_{\text{top+thin+bot}}) - \mathbf{R}_{\text{bot}}(1 + \mathbf{Q}_{\text{top+bot}})$$

$$= \mathbf{R}_{\text{thin+bot}}(1 + \mathbf{Q}_{\text{top+bot}} + \mathbf{Q}_{\text{top+thin+bot}} - \mathbf{Q}_{\text{top+bot}}) - \mathbf{R}_{\text{bot}}(1 + \mathbf{Q}_{\text{top+bot}})$$

$$= \mathbf{R}_{\text{thin+bot}}(1 + \mathbf{Q}_{\text{top+bot}}) + \mathbf{R}_{\text{thin+bot}}(\mathbf{Q}_{\text{top+thin+bot}} - \mathbf{Q}_{\text{top+bot}}) - \mathbf{R}_{\text{bot}}(1 + \mathbf{Q}_{\text{top+bot}})$$

$$= (\mathbf{R}_{\text{thin+bot}} - \mathbf{R}_{\text{bot}})(1 + \mathbf{Q}_{\text{top+bot}}) + \mathbf{R}_{\text{thin+bot}}(\mathbf{Q}_{\text{top+thin+bot}} - \mathbf{Q}_{\text{top+bot}})$$

$$= (\mathbf{R}_{\text{thin+bot}} - \mathbf{R}_{\text{bot}})(1 + \mathbf{Q}_{\text{top+bot}}) + \mathbf{R}_{\text{thin+bot}}(\mathbf{R}^*_{\text{top}}\mathbf{R}_{\text{thin+bot}} - \mathbf{R}^*_{\text{top}}\mathbf{R}_{\text{bot}})$$

$$= (\mathbf{R}_{\text{thin+bot}} - \mathbf{R}_{\text{bot}})(1 + \mathbf{Q}_{\text{top+bot}}) + \mathbf{R}_{\text{thin+bot}}\mathbf{R}^*_{\text{top}}(\mathbf{R}_{\text{thin+bot}} - \mathbf{R}_{\text{bot}})$$

$$= (\mathbf{R}_{\text{thin+bot}} - \mathbf{R}_{\text{bot}})(1 + \mathbf{Q}_{\text{top+bot}}) + (\mathbf{R}_{\text{thin+bot}} - \mathbf{R}_{\text{bot}} + \mathbf{R}_{\text{bot}})\mathbf{R}^*_{\text{top}}(\mathbf{R}_{\text{thin+bot}} - \mathbf{R}_{\text{bot}})$$

$$= (\mathbf{R}_{\text{thin+bot}} - \mathbf{R}_{\text{bot}})(1 + \mathbf{Q}_{\text{top+bot}}) + \mathbf{R}_{\text{bot}}\mathbf{R}^*_{\text{top}}(\mathbf{R}_{\text{thin+bot}} - \mathbf{R}_{\text{bot}}) + (\mathbf{R}_{\text{thin+bot}} - \mathbf{R}_{\text{bot}})\mathbf{R}^*_{\text{top}}(\mathbf{R}_{\text{thin+bot}} - \mathbf{R}_{\text{bot}})$$

$$= (\mathbf{R}_{\text{thin+bot}} - \mathbf{R}_{\text{bot}})(1 + \mathbf{Q}_{\text{top+bot}}) + \mathbf{Q}^*_{\text{top+bot}}(\mathbf{R}_{\text{thin+bot}} - \mathbf{R}_{\text{bot}})$$

$$= (1 + \mathbf{Q}^*_{\text{top+bot}})(\mathbf{R}_{\text{thin+bot}} - \mathbf{R}_{\text{bot}})(1 + \mathbf{Q}_{\text{top+bot}}). \tag{A12}$$

In step 3, $\mathbf{Q}_{\text{top+thin+bot}}$ and $\mathbf{Q}_{\text{top+bot}}$ are replaced by the first term of their expansions; in step 7, the last term is removed because

it depends on $(dz)^2$; in step 8, $\mathbf{R}_{\text{bot}}\mathbf{R}^*_{\text{top}}$ is approximated by $\mathbf{Q}^*_{\text{top+bot}}$. Because $\mathbf{R}_{\text{thin+bot}} - \mathbf{R}_{\text{bot}}$ is small, step 8 only introduces a small error. In order to get a compact form, in step 9 a small term, $\mathbf{Q}^*_{\text{top+bot}}(\mathbf{R}_{\text{thin+bot}} - \mathbf{R}_{\text{bot}})\mathbf{Q}_{\text{top+bot}}$, is added.

Inserting Eq. A12 into Eq. A11 we obtain the linearized form of the difference $d\mathbf{R}$:

$$d\mathbf{R} = \mathbf{R}_{\text{top+thin+bot}} - \mathbf{R}_{\text{top+bot}} = (\mathbf{E}_{\text{top}} + \mathbf{T}^*_{\text{top}})\left[(1 + \mathbf{Q}^*_{\text{top+bot}})(\mathbf{R}_{\text{thin+bot}} - \mathbf{R}_{\text{bot}})(1 + \mathbf{Q}_{\text{top+bot}})\right](\mathbf{E}_{\text{top}} + \mathbf{T}_{\text{top}}). \tag{A13}$$





Here the term $(\mathbf{R}_{\text{thin+bot}} - \mathbf{R}_{\text{bot}})$ contains the optical properties of the thin layer: $k_{\text{sca}}$, $k_{\text{ext}}$ and $\mathbf{Z}$. This term can be written in terms of these optical properties by using Eq. A8 and inserting the thin-layer formulas Eqs. 1–5:

$$\mathbf{R}_{\text{thin+bot}} - \mathbf{R}_{\text{bot}} = \mathbf{R}_{\text{thin}} + [\mathbf{T}^*_{\text{thin}} - (1 - \mathbf{E}_{\text{thin}})]\mathbf{R}_{\text{bot}} + \mathbf{R}_{\text{bot}}[\mathbf{T}_{\text{thin}} - (1 - \mathbf{E}_{\text{thin}})] + \mathbf{R}_{\text{bot}}\mathbf{T}^*_{\text{thin}}\mathbf{R}_{\text{bot}} \tag{A14}$$

$$= k_{\text{sca}}(z)\mathbf{Z}'_{-+}(z)dz$$
$$+ [k_{\text{sca}}(z)\mathbf{Z}'_{--}(z) - k_{\text{ext}}(z)(1/\mu)]\mathbf{R}_{\text{bot}}dz$$
$$+ \mathbf{R}_{\text{bot}}[k_{\text{sca}}(z)\mathbf{Z}'_{++}(z) - k_{\text{ext}}(z)(1/\mu)]dz$$
$$+ \mathbf{R}_{\text{bot}}[k_{\text{sca}}(z)\mathbf{Z}'_{+-}(z)]\mathbf{R}_{\text{bot}}dz$$

$$= \mathbf{W}(z)dz. \tag{A15}$$

(For the definition of the phase matrix $\mathbf{Z}'$ with special supermatrix weights we refer to De Haan (2012); De Haan et al. (1987)). If we now insert this relation, Eq. A15, into Eq. A13, we get the end result:

$$d\mathbf{R} = [\mathbf{E}_{\text{top}} + \mathbf{T}^*_{\text{top}}](1 + \mathbf{Q}^*_{\text{top+bot}})\mathbf{W}(z)(1 + \mathbf{Q}_{\text{top+bot}})[\mathbf{E}_{\text{top}} + \mathbf{T}_{\text{top}}]dz \tag{A16}$$

$$\mathbf{Q}^*_{\text{top+bot}} = \mathbf{R}_{\text{bot}}\mathbf{R}^*_{\text{top}} + \mathbf{R}_{\text{bot}}\mathbf{R}^*_{\text{top}}\mathbf{R}_{\text{bot}}\mathbf{R}^*_{\text{top}} + ... \tag{A17}$$

This formula, Eq. A16, is the derivative equation given in Eq. 15.

## Appendix B: Pseudo-spherical correction

The pseudo-spherical correction in DISAMAR corrects for the curvature of the atmosphere for incident sunlight. For the other light paths the atmosphere remains plane-parallel. In order to calculate the pseudo-spherical correction, the adding scheme for illumination of the layers from below is implemented. These equations are (Eqs. B1–B5):

$$\mathbf{Q}^*_{k,k+1} = \mathbf{R}_k\mathbf{R}^*_{k+1} + \mathbf{R}_k\mathbf{R}^*_{k+1}\mathbf{R}_k\mathbf{R}^*_{k+1} + ... \tag{B1}$$

$$\mathbf{D}^*_{k,k+1} = \mathbf{T}^*_k + \mathbf{Q}^*_{k,k+1}\mathbf{E}_k + \mathbf{Q}^*_{k,k+1}\mathbf{T}^*_k \tag{B2}$$

$$\mathbf{U}^*_{k,k+1} = \mathbf{R}^*_{k+1}\mathbf{E}_k + \mathbf{R}^*_{k+1}\mathbf{D}^*_{k,k+1} \tag{B3}$$

$$\mathbf{R}^*_{k,k+1} = \mathbf{R}^*_k + \mathbf{E}_k\mathbf{U}^*_{k,k+1} + \mathbf{T}_k\mathbf{U}^*_{k,k+1} \tag{B4}$$

$$\mathbf{T}^*_{k,k+1} = \mathbf{T}^*_{k+1}\mathbf{E}_k + \mathbf{T}^*_{k+1}\mathbf{D}^*_{k,k+1} + \mathbf{E}_{k+1}\mathbf{D}^*_{k,k+1}. \tag{B5}$$

We assume that the individual layers remain plane-parallel – the only correction that is needed for the reflectance is to change the attenuation terms for incident sunlight. For the adding method it means replacing $\mathbf{E}_{k+1}$ – when it occurs on the right-hand side of a term – by a new attenuation term. The pseudo-spherical correction method follows Spurr (2002), so the details are not given here.



The partial derivative of the reflectance to the absorption coefficient, Eq. 25 from Sect. 3.3.4, can be rewritten as:

$$\frac{\partial^2 \mathbf{R}}{\partial k_{\mathrm{abs}} \partial z} = -[\mathbf{E}_{\mathrm{top}} + \Delta_3 \tilde{\mathbf{D}} \Delta_3][\mathbf{1}/\boldsymbol{\mu}]\mathbf{U}$$
$$- \Delta_3 \tilde{\mathbf{U}} \Delta_3 [\mathbf{1}/\boldsymbol{\mu}]\mathbf{D}$$
$$- \Delta_3 \tilde{\mathbf{U}} \Delta_3 [\mathbf{1}/\boldsymbol{\mu}]\mathbf{E}_{\mathrm{top}}. \tag{B6}$$

With the pseudo-spherical correction, the last term at the right-hand side of Eq. B6 should be replaced by Eq. B9. Equations B7–B9 present the derivation of the pseudo-spherical correction:

$$\Delta_3 \tilde{\mathbf{U}}_k(\mu_0, \mu) \Delta_3 [\mathbf{1}/\boldsymbol{\mu}]\mathbf{E}_{\mathrm{top}}(\mu_0) = \left\{ \sum_{l=0}^{k} \exp\left(\frac{\tau_k - \tau_l}{\mu_0}\right) \left( \sum_{n=1}^{\infty} \mathbf{U}_l^{\mathrm{local},n}(\mu_0, \mu) \right) \right\} \frac{1}{\mu_0} \exp\left(-\frac{\tau_k}{\mu_0}\right) \tag{B7}$$

$$\Delta_3 \tilde{\mathbf{U}}_k(\mu_0, \mu) \Delta_3 [\mathbf{1}/\boldsymbol{\mu}]\mathbf{E}_{\mathrm{top}}(\mu_0) = \left\{ \sum_{l=0}^{k} \left( \sum_{n=1}^{\infty} \mathbf{U}_l^{\mathrm{local},n}(\mu_0, \mu) \right) \right\} \frac{1}{\mu_0} \exp\left(-\frac{\tau_l}{\mu_0}\right) \tag{B8}$$


$$\Delta_3 \tilde{\mathbf{U}}_k(\mu_0, \mu) \Delta_3 [\mathbf{1}/\boldsymbol{\mu}]\mathbf{E}_{\mathrm{top}}^{\mathrm{sph}}(\mu_0) = \left\{ \sum_{l=0}^{k} \left( \sum_{n=1}^{\infty} \mathbf{U}_l^{\mathrm{local},n}(\mu_0, \mu) \right) \right\} \frac{(R_E + z_k) \exp(-\tau_l^{\mathrm{slant}}(\mu_0))}{\sqrt{(R_E + z_k)^2 - (R_E + z_l)^2(1 - \mu_0^2)}} \tag{B9}$$

where $R_E$ is the radius of the Earth in km, and $z_k$ and $z_l$ are the altitudes of levels $k$ and $l$, respectively, in km. Here we use the local value of $\mathbf{U}$ to distinguish between the different levels below level $k$. The factor $\exp((\tau_k - \tau_l)/\mu_0)$ occurs in Eq. B7, because we used the transpose of $\mathbf{U}$; that is also the reason for the order of the arguments for $\mathbf{U}_l^{\mathrm{local},n}$. The sum over $n$ is the
sum over the orders of scattering. For a plane-parallel atmosphere Eq. B7 can be written as Eq. B8. For a spherical atmosphere, $\exp(-\tau_l/\mu_0)$ should be replaced by $\exp(-\tau_l^{\mathrm{slant}}(\mu_0))$, noted as $\mathbf{E}_{\mathrm{top}}^{\mathrm{sph}}$ in Eq. B10, and the factor $1/\mu_0$ should be replaced by $1/\cos\theta_0'$, where $1/\cos\theta_0' = \frac{(R_E + z_k)}{\sqrt{(R_E + z_k)^2 - (R_E + z_l)^2(1 - \mu_0^2)}}$.

The partial derivative of the reflectance to the scattering coefficient, Eq. 26 from Sect. 3.3.4, becomes with the pseudo-spherical correction:

$$\frac{\partial^2 \mathbf{R}}{\partial k_{\mathrm{sca}} \partial z} = [\mathbf{E}_{\mathrm{top}} + \Delta_3 \tilde{\mathbf{D}} \Delta_3]\mathbf{Z}'_{-+}(z)[\mathbf{E}_{\mathrm{top}}^{\mathrm{sph}} + \mathbf{D}]$$
$$+ [\mathbf{E}_{\mathrm{top}} + \Delta_3 \tilde{\mathbf{D}} \Delta_3]\mathbf{Z}'_{--}(z)\mathbf{U}$$
$$+ \Delta_3 \tilde{\mathbf{U}} \Delta_3 \mathbf{Z}'_{++}(z)[\mathbf{E}_{\mathrm{top}}^{\mathrm{sph}} + \mathbf{D}]$$
$$+ \Delta_3 \tilde{\mathbf{U}} \Delta_3 \mathbf{Z}'_{+-}(z)\mathbf{U}$$
$$+ \frac{\partial^2 \mathbf{R}}{\partial k_{\mathrm{abs}} \partial z}. \tag{B10}$$

This completes the formulae for the pseudo-spherical correction of the derivatives.



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
