# Peer review of "Introduction of the DISAMAR radiative transfer model: Determining Instrument Specifications and Analysing Methods for Atmospheric Retrieval (version 4.1.5)"

_Geoscientific Model Development, 2021_

## Author Comment (AC1)

We would like to thank Dr. Zhai for the comments and suggestions on the manuscript. We have replied to all and revised the manuscript. Please see the replies below. The original comments are in black italic font, and the replies to the comments are in blue normal font.

*This paper describe the Determining Instrument Specifications and Analysing Methods for Atmospheric Retrieval (DISAMAR) computer code, which performs both forward radiative transfer model and inversion simulations for the Earth's atmosphere with different components, for instance, trace gases, aerosols, and clouds, and properties of the ground surface from passive remote sensing observations of the Earth. The main novelty is that DISAMAR flexibly uses a variety of radiative transfer methods when solving multiple scattering of light in the atmosphere, including the layer-based orders of scattering method, adding and doubling, integration of source function, etc. For weakly gas absorbing spectral region, the DISMAS (DIfferential and SMooth Absorption Separated) method is developed to significantly expedite the simulation time while keeping the accuracy. Jacobian (differential of radiance field with respect to retrieval parameters) can be calculated semi-analytically, which is a great advantage in comparison with the finite difference method. The manuscript is clearly written and organization is logical. I suggest the publication of this paper at Geoscientific Model Development (GMD) with some minor revisions. Specifically:*

1. *Figure 1 and 2 are referenced out of order. I suggest the authors revise the manuscript to reference them in sequence. Minimally, they could simply rename Fig. 1 as Fig. 2 and vice versa.*

We have arranged the figures in the right order. In the revised manuscript, Fig. 1 becomes Fig. 3.

2. *Figure 2, the selection of wavelength grids is quite vague to me, especially when absorbing lines are involved in the channel. I strongly recommend the author revise the description the wavelength selection scheme.*

The determination of the wavelength grid has two steps: selection of the wavelength interval and division of the interval using Gaussian points. We have revised the description of the wavelength grid in 3.2.1 close to line 115.

original :
*" For constructing the wavelength grid, we start from the shortest wavelength with the interval of one full-width-half-maximum (FWHM) of the ISRF. If there is a strong absorption line in the interval, the interval is reduced until the position of the strong line is reached, so this interval is smaller than the FWHM. The next interval starts from the position of the strong line. This procedure is repeated until the end of the wavelength range. The number of Gaussian points is scaled with the size of the interval, which means that a smaller interval is having less Gaussian points than the number of Gaussian points specified for the FWHM. Therefore, the wavelength grid is not equidistant: a finer wavelength grid is used for denser absorption lines, and a coarser grid is created if there are no absorption lines."*

new:

There are two steps to construct the high resolution wavelength grid: 1) determining the wavelength intervals and 2) dividing each interval with proper Gaussian division points. The full-width-half-maximum (FWHM) and number of Gaussian division points for one FWHM ($N_0$) and minimum number of Gaussian division points are specified in the configuration file.

1) We start from the shortest wavelength with an interval of one FWHM of the ISRF. If there is a strong absorption line in the interval, the boundary of the interval is set to the position of the strong line, so this interval is smaller than one FWHM. The next interval starts from the position of the strong line with one FWHM interval again. This process is repeated until the last wavelength of the wavelength grid.

2) If the interval is one FWHM, the interval is divided by the number of Gaussian points. If the interval is smaller than one FWHM, the number of Gaussian points is scaled with the size of the interval. For example, if the interval is half of one FWHM, the number of Gaussian point is 0.5 $N_0$. If the scaled number of Gaussian points is smaller than the minimum number of Gaussian points, the minimum number of Gaussian points is used. Note that the Gaussian quadrature weights and abscissae are determined for each interval.

Therefore, the wavelength grid is not equidistant: a finer wavelength grid is used for denser absorption lines, and a coarser grid is created if there are no absorption lines.

3. *Figure 3, I thought it would be more natural to use optical depth as a vertical coordinate in radiative transfer. Thus to me using pressure as the vertical coordinates is a bit unusual. This is just a comment and I won't force the authors to make any changes, as this would be most likely an overhaul of the computer code.*

Thank you for this comment. We think that pressure as a vertical coordinate is convenient for users and applications, therefore it is used in the input and output of DISAMAR. However, we calculate optical thickness per layer, because the optical thickness is the vertical coordinate that is used in the doubling-adding scheme.

4. *Line 150-151, the paper discussed different Gaussian quadrature points for different optical depth situation. Again it would much natural to use optical depth as the vertical grid, so that you would easily built a universal criterion of how many discrete layers are needed in terms of optical depth. By the way, how large is the optical depth considered as "thick"?*

We do not have a universal criterion to determine number of layers. We have to try out different settings. We usually assume optical thickness of larger than 10 is 'thick'.

5. *Line 408, please give a list of "strong" absorbers and their associated wavelength ranges to which DISMAS should not be applied. How strong of a gas absorption line is considered strong?*

Weak absorbers such as $NO_2$, HCHO, $O_3$ (visible only), BrO, $SO_2$, $O_2$-$O_2$ can be used in DISMAS. The optical thickness of these weak absorbers is smaller than 0.1, which meets our assumption. $O_2$, $H_2O$, $CH_4$, CO, $O_3$ (UV) cannot be used in DISMAS. We have added the list of absorbers that can be used in DISMAS in the manuscript close to line 425.

DISMAS can be used for the retrieval of total columns of weakly absorbing gases, like $O_3$ (in visible wavelength range), $NO_2$, $SO_2$, BrO, and $CH_2O$, but not for strong absorbers, such as $O_2$, $H_2O$, $CH_4$, CO, $O_3$ (UV).

*6. For the spherical shell correction, there are some new developments recently. Specifically Korkin, E.-S. Yang, R. Spurr, C. Emde, P. Zhai, N. Krotkov, A. Vasilkov, A. Lyapustin,*

*Numerical results for polarized light scattering in a spherical atmosphere, Journal of Quantitative Spectroscopy and Radiative Transfer, Volume 287, 2022, 108194, ISSN 0022-4073, https://doi.org/10.1016/j.jqsrt.2022.108194.*

*(https://www.sciencedirect.com/science/article/pii/S0022407322001297)*

*Peng-Wang Zhai, Yongxiang Hu, An improved pseudo spherical shell algorithm for vector radiative transfer, Journal of Quantitative Spectroscopy and Radiative Transfer,*

*Volume 282, 2022, 108132, ISSN 0022-4073, https://doi.org/10.1016/j.jqsrt.2022.108132.*

*([https://www.sciencedirect.com/science/article/pii/S0022407322000693](https://www.sciencedirect.com/science/article/pii/S0022407322000693))*

Thank you for the references.

*For Raman scattering and other inelastic scattering in the ocean waters, there are some new development as well:*

*Peng-Wang Zhai, Yongxiang Hu, David M. Winker, Bryan A. Franz, and Emmanuel Boss, "Contribution of Raman scattering to polarized radiation field in ocean waters," Opt. Express 23, 23582-23596 (2015)*

*Peng-Wang Zhai, Yongxiang Hu, David M. Winker, Bryan A. Franz, Jeremy Werdell, and Emmanuel Boss, "Vector radiative transfer model for coupled atmosphere and ocean systems including inelastic sources in ocean waters," Opt. Express 25, A223-A239 (2017)*

Thank you for the references.

---

## Author Comment (AC2)

We much appreciate that Dr. Duan spent time on reading the manuscript and giving constructive comments. We have replied to all comments and revised the manuscript. Please see the replies below. The original comments are in black italic font, and the replies to the comments are in blue normal font.

*A forward modelling is essential for understanding the physics behind radiation measurements, it is also the base of building accurate algorithm for remote sensing, DISAMAR is of a such important tool for retrieving atmospheric gases and particles, and it has been used for TROPOMI and SP4/SP5 observations. This model introduces several numerical techniques such as layer-based SOS method, semi-analytical derivatives etc. to improve its computation efficiency as well as its computational accuracy. I agree its publication in GMD after minor revision, and I'm looking forward its application for more space-born instruments in future.*

1. *Around line 280, it is not clearly to see how can we get EQ. 25 from EQ 24, please add more description.*

We have added some explanations close to line 298 as follows:

First, replace $k_{ext}$ with $k_{sca} + k_{abs}$ in Eq. 24 and separate terms having $k_{abs}$ and $k_{sca}$. Then calculate partial derivative of the reflectance w.r.t. $k_{abs}$ and $k_{sca}$, respectively. All $k_{sca}$ terms disappear in Eq. 25, while Eq. 26 has no $k_{abs}$ terms.

2. *For the convolution of the ISRF around Line 109. "If there is a strong absorption line in the interval, the interval is reduced until the position of the strong line is reached, so this interval is smaller than the FWHM", this is easy to do for O2-A band which is regular spaced, for irregular spaced absorption band, say, water vapor band, is this method easy to apply this algorithm?*

The method is applied to all absorption bands, including water vapor bands. The interval in DISAMAR is irregular, so it does not matter whether the absorption band is regular or irregular spaced. We have revised the description of the wavelength grid, see lines 115-125 in the revised manuscript. It is easier to understand now.

3. *Line 114,"Typically the number of Gaussian division points is between 3 and 30 per wavelength interval in the O2 A-band for a FWHM of 0.5 nm." Is there a rule for the reader to know how to choose the number between 3 and 30.*

We do not have a rule for the reader to choose the number of Gaussian division points. The choice of the Gaussian points is based on experiences. It is a combination of accuracy and computation time. We have added more explanations on the construction of the wavelength grid in the revised version.

4. *"only the adding of different layers and the subsequent calculation of the internal field is replaced by the successive orders of scattering method." Please add several sentences to make clear how to calculate the internal field?*

The text close to line 348 has been revised as follows:

Only the adding of different layers and the subsequent calculation of the internal field is replaced by the successive orders of scattering method (see Eqs. 35 – 38).

Added text close to line 359:

In order to calculate the total internal fields U and D (see Eqs. 33-38), we first need to calculate local internal fields $U^{local}$ and $D^{local}$. (see Eqs. 31-32).

---

## Author Comment (AC3)

We would like to thank Dr. Stegmann for the detailed comments and suggestions. We have replied to all comments and revised the manuscript. Please see the replies below. The original comments are in black italic font, and the replies to the comments are in blue normal font.

*Journal: Geophysical Model Development*

*Year: 2022*

*Title: Introduction of the DISAMAR radiative transfer model: Determining Instrument Specifications and Analysing Methods for Atmospheric Retrieval (version 4.1.5)*

*Comments:*

*In the manuscript, the authors describe the so-called DISAMAR one-dimensional radiative transfer model.*

*The model is described by the authors as a polarized all-sky radiative transfer model, i.e. it is suitable for purely absorbing clear-sky atmospheres and atmospheres with scattering clouds.*

*The application focus of said model are satellite radiance retrievals, in particular for the TROPOMI instrument on board the european Sentinel-5p satellite.*

*Nevertheless, the authors list a suite of different available solvers and a range of additional model features that are not required for retrievals, such as irradiance computations.*

*It is emphasized by the authors that the primary advantage of their model is the seamless combination of all necessary features for satellite remote sensing in their model.*

*Comments on the Introduction:*

*- line 44: The assumption that the atmospheric input profile of the radiative transfer model is hydrostatic is of some importance. How does this approximation impact the DISAMAR retrieval results?*

The hydrostatic assumption is only used when determining the altitude grid. The altitude grid may be slightly different if the input pressure, temperature profile are not hydrostatic. For the actual calculation of the number of molecules we use the pressure and temperature profile and the trace gas mixing ratio profile specified in the input configuration file.
In the revised manuscript we removed this sentence in the introduction and added the explanation of the conversion from pressure grid to altitude grid in Sect. 3.2.2 after the description of the pressure grid. The conversion from pressure grid to altitude grid is now described in Appendix C.

*- line 47: Is a Lambertian reflectance the only surface reflectance type available? Does this limit the accuracy of the model over ocean surfaces where the Cox-Munk model is typically applied?*

Yes, DISAMAR version 1.4.5 has only Lambertian surface reflectance available. Indeed, it means a limitation of DISAMAR to apply it over ocean surface or vegetation with strong BRDF. We may implement the Cox-Munk model if it is required by users.

*- It would be advantageous to provide a list of other relevant radiative transfer models with similar purpose and complexity in comparison to DISAMAR.*

*Examples include the CRTM [1-3] and RTTOV [4].*

Thank you for the suggestion. We included the references and revised the text in the introduction close to line 54. Text added:

Due to the time consuming line-by-line calculations, DISAMAR is not suitable for fast computations or application in numerical weather prediction (NWP) models. We would recommend RTTOV (Saunders et al., 2018) and CRTM (Lu et al., 2021, Karpowicz et al., 2022, Stegmann et al., 2022) as fast radiative transfer models for NWP applications. Actually DISAMAR has been used to benchmark RTTOV simulations in the UV/visible wavelength range.

*Comments on Section 2:*

*- line 65: Please provide a short explanation on the purpose of the wavelength grid.*

To improve the explanation in the previous version of the manuscript, the text has been revised close to line 67 in the revised manuscript.

Old version:

'This makes the integration more accurate than an equidistant grid with similar number of grid points. '

New version:

'This makes the integration over altitude, wavelength, or polar angle more accurate than the integration at an equidistant grid with a similar number of grid points.'

*- lines 71 and 72: Please provide references for the application of the derivatives for optimal estimation and the application to the error covariance matrix, gain vectors, and averaging kernel.*

We added Rodgers (2000) as reference, after line 75.

*- line 73: The formal theory of evaluating the derivatives of a computer program is quite well developed [5].*

Thank you for the reference. We have included it in the paper close to line 77.

In DISAMAR all derivatives are calculated in a semi-analytical manner although algorithmic differentiation can be used to evaluate the derivatives (Griewank and Walther, 2008).

*Could you please elaborate whether your semi-analytical approach computes the forward-mode (tangent-linear) or reverse-mode (adjoint) derivative of your code output/ the radiance spectrum?*

The semi-analytical approach computes the forward-mode derivatives of the radiance spectrum.

We have added this sentence in the paper.  (add the sentence close to line 91)

*Comments on Section 3:*

*- line 92: Are there any restrictions when using a tabulated ISRF? It is stated in Section 2 that the radiance wavelength grid is given on a set of Gaussian quadrature points. Are the tabulated values automatically interpolated onto the grid?*

There is no restriction when using a tabulated ISRF. In fact we have used tabulated GOME-2 and TROPOMI ISRFs. The tabulated ISRF values are interpolated onto the wavelength grid.

We have revised the text close to line 97.

"During the convolution, the wavelength grid of the ISRF is interpolated to a high resolution wavelength grid (see Sect. 3.2.1)."

*- line 106: How does the line-by-line absorption model impact the calculation time of DISAMAR?*

The DISAMAR forward model is relatively slow.

*Does DISAMAR include faster absorption models when calculation time is a constraint?*

There is no faster absorption model in DISAMAR. DISAMAR has been used to train a neural network for aerosol layer height retrieval to improve the speed of the retrieval algorithm.

*- line 140: Please explain how the pressure levels are translated into altitude levels. Are you using the hypsometric equation?*

We use the Hydrostatic equation to convert pressure level to altitude levels.  The complete formulas are added in Appendix C (Conversion from pressure grid to altitude grid) in the revised manuscript. We refer to the formulas in Appendix C close to line  155.

*- line 185: There are different adding-doubling initialization schemes [6] and this is known as the infinitesimal generator initialization.*

Thank you for the reference (*Wiscombe, 1976),* of which we are aware. The choice of initialization schemes has been discussed by De Haan et al. (1987) in the description of the polarized Doubling-Adding algorithm. In the Doubling-Adding code two orders of scattering are used to initialize the doubling scheme. De Haan et al. (1987) did not use the so-called diamond method, because it was doubtful if the diamond method was better than the two-orders-of-scattering method. Also the diamond method was not tested for polarized light in Wiscombe (1976).

DISAMAR uses single scattering instead of two orders of scattering to initialize the doubling scheme, because of speed.

*- I have not checked equations (19) to (26) for correctness.*

We think they are correct, because we have checked the semi-analytical derivatives using the numerical perturbation method.

*- line 335: If the Layer-Based Orders of Scattering method is fast for optically thin clouds, wouldn't it provide some advantages to initialize the Adding-Doubling solver with a LABOS solution, since it spends a lot of computation time doubling a small initial layer?*

Thank you for the suggestion. We only use single scattering to initialize the adding-doubling solver, because it is faster than two orders of scattering that is used in the Doubling Adding method (de Haan et al., 1987; Stammes et al., 1989).

*- Is the LABOS method related to the Successive Order of Scattering approximation? If so, what are the characteristic differences?*

The principle is the same between LABOS and Successive Order of Scattering approximation. It was explained in manuscript in lines 331-335.

"Here one order of scattering represents scattering by an atmospheric layer. This differs from the classical method of successive orders of scattering where the scattering element is a volume-element of the atmosphere instead of a layer (Lenoble et al., 2007; Min and Duan, 2004). In the adding method one deals with matrix-matrix multiplications, whereas in LABOS one deals with matrix-vector multiplications. However, in LABOS the calculations have to be repeated for the different orders of scattering. "

*References:*

*[1] C. H. Lu, Q. Liu, S. Wei, B. T. Johnson, C. Dang et al. (2021): The Aerosol Module in the Community Radiative Transfer Model (v2.2 and v2.3): accounting for aerosol transmittance effects on the radiance observation operator. Geosci. Model Dev., 15, 1317–1329.*

*[2] B. M. Karpowicz, P. G. Stegmann, B. T. Johnson, H. W. Christopherson et al. (2022): pyCRTM: At python interface for the community radiative transfer model. J. Quant. Spec. Rad. Trans. 288.*

*[3] P. G. Stegmann, B. T. Johnson, I. Moradi, B. Karpowicz, W. McCarty (2022): A deep learning approach to fast radiative transfer. J. Quant. Spec. Rad. Trans. 280.*

*[4] R. Saunders, J. Hocking, E. Turner, P. Rayer, D. Rundle, P. Brunel, et al. (2018): An update on the RTTOV fast radiative transfer model (currently at version 12). Geosci. Model Dev., 11, 2717–2737*

*[5] A. Griewank, A. Walther: Evaluating Derivatives, Principles and Techniques of Algorithmic Differentiation. Society for Industrial and Applied Mathematics; 2nd edition (November 6, 2008)*

*[6] Wiscombe, W. J., 1976. On initialization, error and flux conservation in the doubling method. J. Quant. Spectrosc. Radiat. Transfer 16, 637-658.*